# SAC1 degrades its lipid substrate PtdIns4*P* in the endoplasmic reticulum to maintain a steep chemical gradient with donor membranes

James P Zewe, Rachel C Wills, Sahana Sangappa, Brady D Goulden, Gerald RV Hammond*

Department of Cell Biology, University of Pittsburgh School of Medicine, Pittsburgh, United States

**Abstract** Gradients of PtdIns4*P* between organelle membranes and the endoplasmic reticulum (ER) are thought to drive counter-transport of other lipids via non-vesicular traffic. This novel pathway requires the SAC1 phosphatase to degrade PtdIns4*P* in a 'cis' configuration at the ER to maintain the gradient. However, SAC1 has also been proposed to act in 'trans' at membrane contact sites, which could oppose lipid traffic. It is therefore crucial to determine which mode SAC1 uses in living cells. We report that acute inhibition of SAC1 causes accumulation of PtdIns4*P* in the ER, that SAC1 does not enrich at membrane contact sites, and that SAC1 has little activity in 'trans', unless a linker is added between its ER-anchored and catalytic domains. The data reveal an obligate 'cis' activity of SAC1, supporting its role in non-vesicular lipid traffic and implicating lipid traffic more broadly in inositol lipid homeostasis and function.

DOI: https://doi.org/10.7554/eLife.35588.001

*For correspondence:
ghammond@pitt.edu

Competing interests: The authors declare that no competing interests exist.

## Introduction

Phosphatidylinositol-4-phosphate (PtdIns4*P*) is arguably the most functionally diverse lipid molecule in eukaryotic cells. Firstly, PtdIns4*P* is a crucial metabolic intermediate in the synthesis of the plasma membrane inositol lipids PtdIns(4,5)$P_2$, PtdIns$P_3$ and PtdIns(3,4)$P_2$ (*Brockerhoff and Ballou, 1962*; *Stephens et al., 1991*; *Posor et al., 2013*), each with their own array of cellular functions (reviewed in [*Balla, 2013*]). Secondly, PtdIns4*P* binds to and thereby recruits and/or activates many proteins involved in cellular traffic (*Tan et al., 2014*). These include proteins regulating vesicular traffic at the endoplasmic reticulum (ER), late endosomes/lysosomes (LEL) and Golgi (*Wang et al., 2007*; *Wang et al., 2003*; *Jović et al., 2012*; *2014*; *Klinkenberg et al., 2014*), as well as non-vesicular lipid transport at the plasma membrane (PM), LEL and Golgi (*Mesmin et al., 2013*; *Chung et al., 2015*; *Moser von Filseck et al., 2015a*; *Zhao and Ridgway, 2017*).

Such a cardinal role in controlling membrane function throughout the secretory and endocytic pathways implies the existence of exquisite homeostatic mechanisms that control PtdIns4*P* abundance. PtdIns4*P* is synthesized by two families of PI 4-kinases, each with their own unique modes of regulation (*Boura and Nencka, 2015*). However, much recent attention has focused on control of PtdIns4*P* through regulation of its degradation. The primary route of the lipid's catabolism is via removal of the 4-phosphate by SAC family lipid phosphatases (*Balla, 2013*). The principle enzyme in budding yeast is the highly conserved SAC1 enzyme (*Guo et al., 1999*; *Rivas et al., 1999*; *Hughes et al., 2000*). SAC1 is an integral membrane protein with two C-terminal transmembrane helices (*Whitters et al., 1993*; *Konrad et al., 2002*; *Nemoto et al., 2000*), which localizes primarily to the ER but is also able to traffic to the Golgi depending on the growth status of the cell

(*Faulhammer et al., 2007*; *Blagoveshchenskaya et al., 2008*). The ER localization at first seemed counter-intuitive, given functions of PtdIns4*P* at the Golgi, PM and endosomes; but the realization that the ER makes extensive membrane contact sites (MCS) with all of these organelles raised a tantalizing possibility as to how ER-localized SAC1 could control PtdIns4*P* abundance: that it could perhaps localize to these MCS and 'reach across' the gap to degrade PtdIns4*P* in a 'trans' configuration (*Phillips and Voeltz, 2016*). Indeed, the crystal structure of SAC1 revealed an approximately 70 amino acid region between the N-terminal catalytic domain and C-terminal transmembrane domains that was disordered in the crystal; this stretch was proposed to be able to span the 15–20 nm gap between ER and organelle at MCS and confer 'trans' activity (*Manford et al., 2010*). Subsequently, targeting of SAC1 to ER-PM MCS by the Osh3 protein was proposed to allow dephosphorylation of PM PtdIns4*P* by cortical ER-localized SAC1 in 'trans', hence controlling plasma membrane inositol lipid synthesis and function (*Stefan et al., 2011*). More recently, dynamic localization of SAC1 to ER-PM MCS has been proposed to regulate PM inositol lipid function in mammalian cells (*Dickson et al., 2016*).

The picture is complicated by a recently proposed novel mode of action for PtdIns4*P*: that its synthesis on cellular membranes can be used to drive counter-transport of other lipids against their concentration gradients (*de Saint-Jean et al., 2011*). In this model, oxysterol binding protein (OSBP)-related lipid transfer proteins operating at MCS collect their lipid cargoes from the ER (where they are synthesized) and transfer them to the destination membrane (such as the PM or Golgi). The lipid transfer domain then avoids a futile reverse transfer reaction because it preferentially collects and back-traffics a PtdIns4*P* molecule instead. Crucially, futile traffic of PtdIns4*P* from the ER back to the destination membrane is prevented because the PtdIns4*P* is degraded by ER-localized SAC1, acting in this case in a 'cis' configuration. Therefore, the transporter must traffic a cargo lipid back to the destination in the next cycle, and the vectorial nature of the transfer is conserved. Ultimately, the energy of ATP hydrolysis by PI4K during PtdIns4*P* synthesis is harnessed to build and maintain a steep chemical gradient of PtdIns4*P* at the destination membrane with respect to the ER. Flow of PtdIns4*P* down this concentration gradient via OSBPs thus powers counter-transport of other lipids against their own chemical gradients. We think of this cycle as a 'phosphoinositide-motive force' (PPInMF), since it is conceptually related to the proton-motive force and other ionic gradients that drive counter-transport of ions and small solutes across membranes (*Mesmin et al., 2013*). Direct evidence for PPInMF-driven transfer reactions have now been presented at the trans-Golgi network (TGN) for sterols (*Mesmin et al., 2013*; *von Filseck et al., 2015b*) and at the PM for phosphatidyl-serine (*Moser von Filseck et al., 2015a*; *Chung et al., 2015*).

A critical requirement for the PPInMF is that SAC1 acts in a 'cis' configuration in the ER; 'trans' activity would act to dissipate the PtdIns4*P* gradients at MCS and make counter-transport much less efficient. In vitro experiments have demonstrated that SAC1 can indeed act in the required 'cis' configuration (*Mesmin et al., 2013*; *von Filseck et al., 2015b*). However, an obligate 'cis' acting SAC1 makes it much harder to understand how PtdIns4*P* abundance is regulated with respect to its other metabolic and trafficking functions, especially at the PM. Therefore, a central question in understanding the regulation of protein and lipid traffic is whether SAC1 activity in cells occurs in an obligatory 'cis' or 'trans' configuration, or whether the enzyme can switch between modes. We address this crucial question here. We present evidence for 'cis' activity in mammalian cells, show that SAC1 fails to enrich at ER-PM MCS and, finally, we show that SAC1 does not possess a conformation that allows it to traverse ER-PM MCS and act in 'trans'. Collectively, our results support a central role for SAC1 in driving the PPInMF, and implicate non-vesicular PtdIns4*P* traffic in the control of inositol lipid metabolism and function more generally.

## Results

### Evidence for a 'cis' acting SAC1

Loss of SAC1 'trans' activity would cause accumulation of its PtdIns4*P* substrate in membranes like the PM and Golgi where the lipid is synthesized, whereas loss of 'cis' activity predicts accumulation of PtdIns4*P* in the ER (*Figure 1D*). *Saccharomyces cerevisiae* with deletions of their *Sac1* gene show 6–10 fold increases in PtdIns4*P* mass (*Rivas et al., 1999*; *Hughes et al., 2000*; *Guo et al., 1999*), with PtdIns4*P* reported at both the PM (*Roy and Levine, 2004*; *Stefan et al., 2011*) and the ER

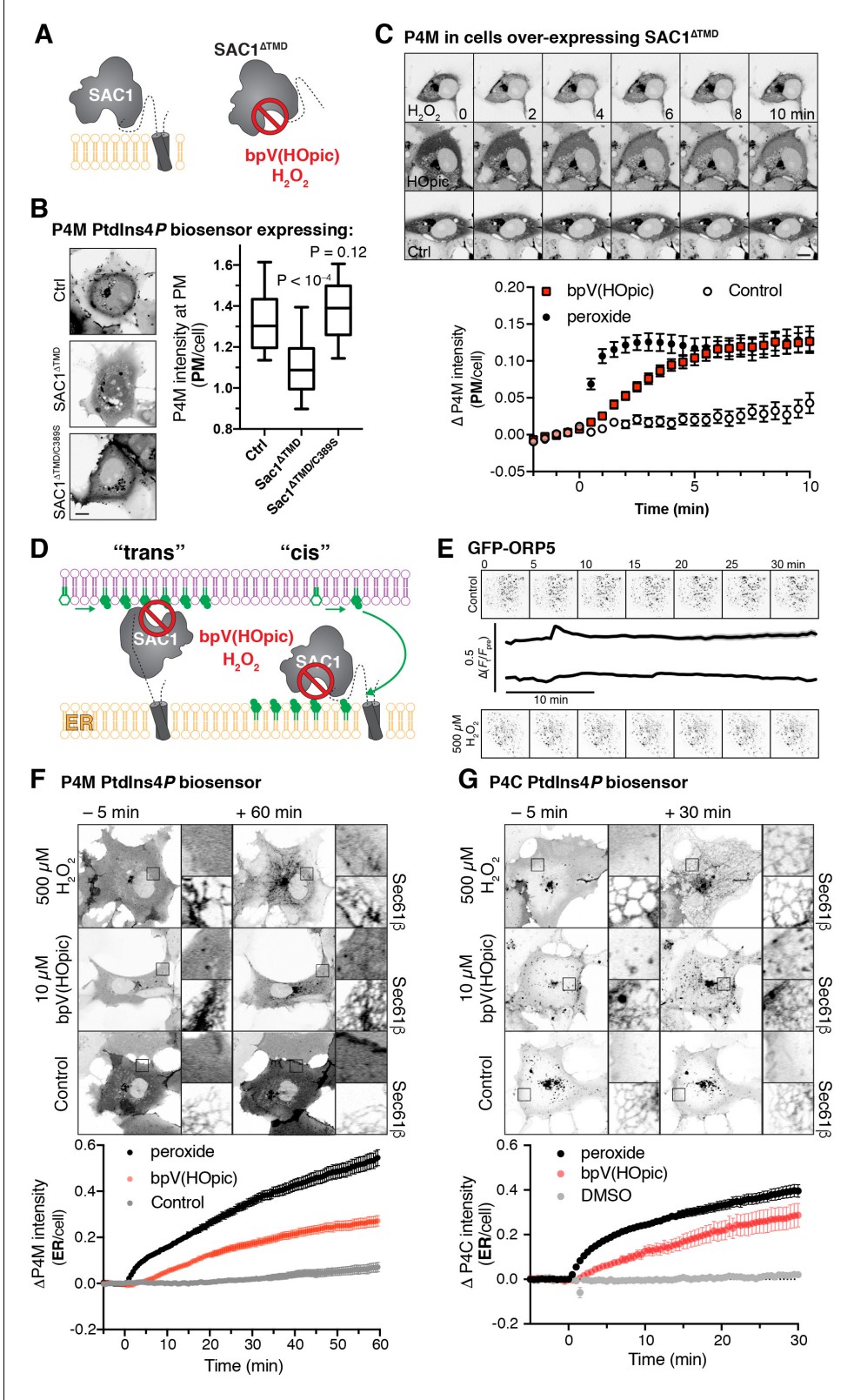

**Figure 1.** Inhibition of SAC1 causes PtdIns4P accumulation in the ER. (**A**) A soluble fragment of SAC1 (SAC1$^{\Delta TMD}$) is inhibited by peroxide and bpV (HOpic). (**B**) SAC1 expression depletes PM PtdIns4P. COS-7 cells transfected with GFP-P4M and either FKBP-mCherry (Ctrl), SAC1$^{\Delta TMD}$-FKBP-mCherry or catalytically inactive SAC1$^{\Delta TMD/C389S}$-FKBP-mCherry were imaged live by confocal microscopy. Representative images are shown (bar = 10 μm). The graph shows P4M intensity at the plasma membrane (defined by CellMask deep red dye) normalized to total cell intensity; box and whisker plot shows
*Figure 1 continued on next page*

*Figure 1 continued*

quartiles and 5–95 percentiles of 90 cells from three independent experiments. *P* values derive from Dunn's multiple comparison test compared to Ctrl after a Kruskal-Wallis test (p<10$^{-4}$). (**C**) Peroxide and bpV(HOpic) inhibit SAC1 in live cells. COS-7 cells were transfected with P4M and SAC1$^{\Delta TMD}$ as in B and imaged by time-lapse confocal microscopy. 500 µM peroxide or 10 µM bpV(HOpic) were added at time 0. P4M intensity was quantified as in B. Data are means ± s.e. of 44 or 45 cells from four independent experiments. Scale bar = 10 µm. (**D**) Predicted PtdIns4*P* accumulation for 'cis' and 'trans' operation of SAC1. (**E**) Peroxide does not disrupt ORP5 localization at ER-PM MCS. Images show TIRF images of COS-7 cells expressing GFP-ORP5 at the indicated times. Traces are means with s.e. shaded for 31–32 cells from three independent experiments. (**F–G**) SAC1 inhibitors cause PtdIns4*P* accumulation in the ER. Time-lapse images of representative COS-7 cells expressing GFP-P4M (**F**) or GFP-P4C (**G**) and treated with inhibitors at time 0. The insets are 10 µm squares, and are expanded at right and show PtdIns4*P* accumulation relative to a co-expressed ER marker, iRFP-Sec61β. Graphs show P4M intensity at the ER (defined by iRFP-Sec61β) normalized to total cell intensity; data are means ± s.e. of 38–41 (**F**) or 29–30 (**G**) cells from three (**G**) or four (**F**) independent experiments.

DOI: https://doi.org/10.7554/eLife.35588.002

The following source data is available for figure 1:

**Source data 1.** Data for panel 1B.
DOI: https://doi.org/10.7554/eLife.35588.003
**Source data 2.** Data for panel 1C.
DOI: https://doi.org/10.7554/eLife.35588.004
**Source data 3.** Data for panel 1E.
DOI: https://doi.org/10.7554/eLife.35588.005
**Source data 4.** Data for panel 1F.
DOI: https://doi.org/10.7554/eLife.35588.006
**Source data 5.** Data for panel 1G.
DOI: https://doi.org/10.7554/eLife.35588.007

(*Roy and Levine, 2004*; *Tahirovic et al., 2005*; *Cai et al., 2014*), depending on the probe used. RNAi of SAC1 in mammalian cells causes 1–2 fold accumulation of PtdIns4*P* (*Cheong et al., 2010*; *Dickson et al., 2016*; *Goto et al., 2016*), with accumulation reported in the ER (*Cheong et al., 2010*; *Blagoveshchenskaya et al., 2008*). On the other hand, acute knock-out of SAC1 in HeLa cells with CRISPR/Cas9 was reported to induce PtdIns4*P* accumulation on the PM and endosomes (*Dong et al., 2016*). However, these experiments are hard to interpret, since the *SACM1L* gene is essential to the survival of single mammalian cells (*Blomen et al., 2015*; *Wang et al., 2015*; *Liu et al., 2008*). Phenotypes in RNAi and knock-out experiments are therefore observed during the rundown of SAC1 protein levels before the cells die. The phenotype observed may thus be exquisitely sensitive to the precise amount of SAC1 protein remaining in the cell at the time of the experiment.

As an alternative approach, we exploited acute chemical inhibition of SAC1. As a member of an especially redox-sensitive family of lipid phosphatases, SAC1 is inherently sensitive to inhibition by oxidizing compounds including bis-peroxovanadates (bpVs) and hydrogen peroxide (*Rosivatz et al., 2006*; *Ross et al., 2007*). In fact, treatment of cells with 500 µM peroxide was shown to induce a massive 7-fold accumulation of PtdIns4*P* in mammalian cells (*Ross et al., 2007*). We therefore sought to determine where such PtdIns4*P* accumulations occur, using our unbiased probe GFP-P4M that detects all cellular pools of PtdIns4*P* (*Hammond and Balla, 2015*; *Hammond et al., 2014*).

Firstly, we wanted to verify that bpVs and peroxide would inhibit SAC1 in the reducing environment of a living cell's cytoplasm (*Figure 1A*). To this end, we over-expressed a soluble SAC1 fragment missing the C-terminal transmembrane domain (ΔTMD). SAC1$^{\Delta TMD}$ expression in COS-7 cells greatly reduced PM localization of P4M, indicating PtdIns4*P* was depleted, whereas a catalytically inactive C389S mutant was without effect (*Figure 1B*). Acute treatment of these cells with 500 µM peroxide led to a rapid (<1 min) recovery of PM PtdIns4*P*, followed by an accumulation of P4M internally (see following paragraph for an explanation); treatment with 10 µM bpV(HOpic) caused a somewhat slower recovery of PM PtdIns4*P* over approximately 5 min (*Figure 1C*). Clearly, the over-expressed SAC1$^{\Delta TMD}$ could be inhibited in the context of a living cell.

Inhibition of SAC1 with these oxidative stress-inducing compounds is likely to induce an ER-stress response, which could potentially lead to disrupted ER-PM MCS (*van Vliet et al., 2017*). However, under our experimental conditions, ER-PM MCS marked with GFP-ORP5 (that would be responsible for PtdIns4*P* traffic) were unaffected by peroxide treatment (*Figure 1E*).

We then sought to inhibit endogenous SAC1 with these compounds. Cells were treated with peroxide or bpV(HOpic) for one hour, which stimulated a rapid (commencing within 5 min) accumulation of ER PtdIns4P with peroxide and a slower, but robust accumulation in the ER with bpV(HOpic) (*Figure 1F*). ER localization of the accumulated PtdIns4P pool was verified by co-expression with iRFP-tagged Sec61β (*Figure 1F*). We observed a similar accumulation of ER-signal with a second, high affinity PtdIns4P biosensor GFP-P4C (*Weber et al., 2014*) as shown in *Figure 1G*. The rapid accumulation at the ER explains the internal accumulation seen in SAC1^ΔTMD over-expressing cells in *Figure 1B*. The most parsimonious explanation for these data is that upon acute inhibition of SAC1, transfer of PtdIns4P to the ER continues, but SAC1 is unable to dephosphorylate it, leading to massive accumulation of the lipid in this compartment (*Ross et al., 2007*). These data are therefore consistent with previous observations (*Cheong et al., 2010*; *Blagoveshchenskaya et al., 2008*) that SAC1 exhibits 'cis' activity in the ER of mammalian cells, though they do not rule out the occurrence of additional 'trans' activity *a priori*.

## SAC1 does not enrich at ER-PM MCS

Dynamic recruitment of SAC1 to ER-PM MCS was recently proposed as a mechanism to modulate 'trans' activity of the enzyme (*Dickson et al., 2016*). Although this is not inconsistent with the firmly established localization of SAC1 throughout the ER (*Rohde et al., 2003*; *Nemoto et al., 2000*), most proteins known to function at MCS are enriched at them too (*Gatta et al., 2017*). Therefore, a clue as to SAC1's preferred mode of activity may be gleaned from a careful analysis of its enrichment (or lack of enrichment) at MCS.

We attempted to use immunofluorescence to localize SAC1, though we failed to identify conditions whereby ER morphology was well enough preserved and specific antibody signal was strong enough to localize SAC1 at high resolution. Instead, we turned to gene editing technology to tag the endogenous SAC1 gene, specifically using a 'split GFP' approach (*Cabantous et al., 2005*; *Leonetti et al., 2016*; *Figure 2A*). We engineered a HEK-293A cell line to stably over-express GFP1-10 (designated 293A^GFP1-10 cells from hereon). We then edited these cells to introduce the GFP11 tag to the N-termini of either the SAC1 protein, the ER-resident Sec61β or the MCS protein Extended Synaptotagmin 1, E-Syt1 (*Giordano et al., 2013*). Genotyping was performed in edited cells using GFP11-specific forward primers and reverse primers corresponding to a ~ 200 bp upstream region in exon 1. In each case, the predicted 200 bp amplicon was produced in the edited cells but not in the un-edited 293A^GFP1-10 cells (*Figure 2B*), though we did observe some longer non-specific PCR products. Sanger sequencing of amplicons spanning the edited site in each case was accomplished with GFP11 primers to verify insertion of the tag in frame.

Confocal imaging revealed the expected ER/Golgi localization of GFP11-SAC1 in edited cells, showing co-localization with the ER marker VAPB and the cis/medial Golgi marker Mannosidase II (*Figure 2C*). Medial confocal optical sections of GFP11-E-Syt1 and GFP-11-Sec61β also revealed exclusively ER localization (*Figure 2C*), consistent with previous reports (*Saheki et al., 2016*; *Leonetti et al., 2016*). To look for ER-PM MCS, we used total internal reflection fluorescence (TIRF) microscopy. Like GFP11-Sec61β, GFP11-SAC1 exhibited a reticular distribution, though unlike Sec61β, it also showed juxta-nuclear enrichment likely corresponding to the bottom of the Golgi (*Figure 2D*). GFP11-E-Syt1 also revealed a reticular distribution by TIRF, but with numerous bright puncta that are most likely ER-PM MCS (*Figure 2D*). Quantitative analysis of the ratio of fluorescence intensity via TIRF imaging (selective for signal within ~100 nm of the coverslip) vs conventional epi-illumination (exciting fluorescence throughout the entire volume of the cell) revealed no enrichment of GFP11-SAC1 close to the basal PM as compared to ER-localized GFP11-Sec61β, whereas GFP11-E-Syt1 showed a marked enrichment (*Figure 2D*). So, it appeared we could not detect enrichment of SAC1 at ER-PM MCS.

To produce a more quantitative, rigorous comparison of MCS vs non-MCS localization of SAC1, we turned to expression of GFP-SAC1 in COS-7 cells, since expressed SAC1 exhibits an identical localization to the endogenous protein (*Rohde et al., 2003*). We expressed GFP-SAC1 along with GFP-ORP5 (*Chung et al., 2015*), GFP-E-Syt2 (*Giordano et al., 2013*) and MAPPER (*Chang et al., 2013*) as positive controls for proteins that localize to ER-PM MCS, along with GFP-calreticulin and GFP-Sec61β as negative controls. For each protein, we co-expressed mCherry-MAPPER as a marker for MCS and iRFP-Sec61β to label the total ER. After acquiring TIRF images of the cells, we background-subtracted the images and compared co-localization of the test protein to both Sec61β and

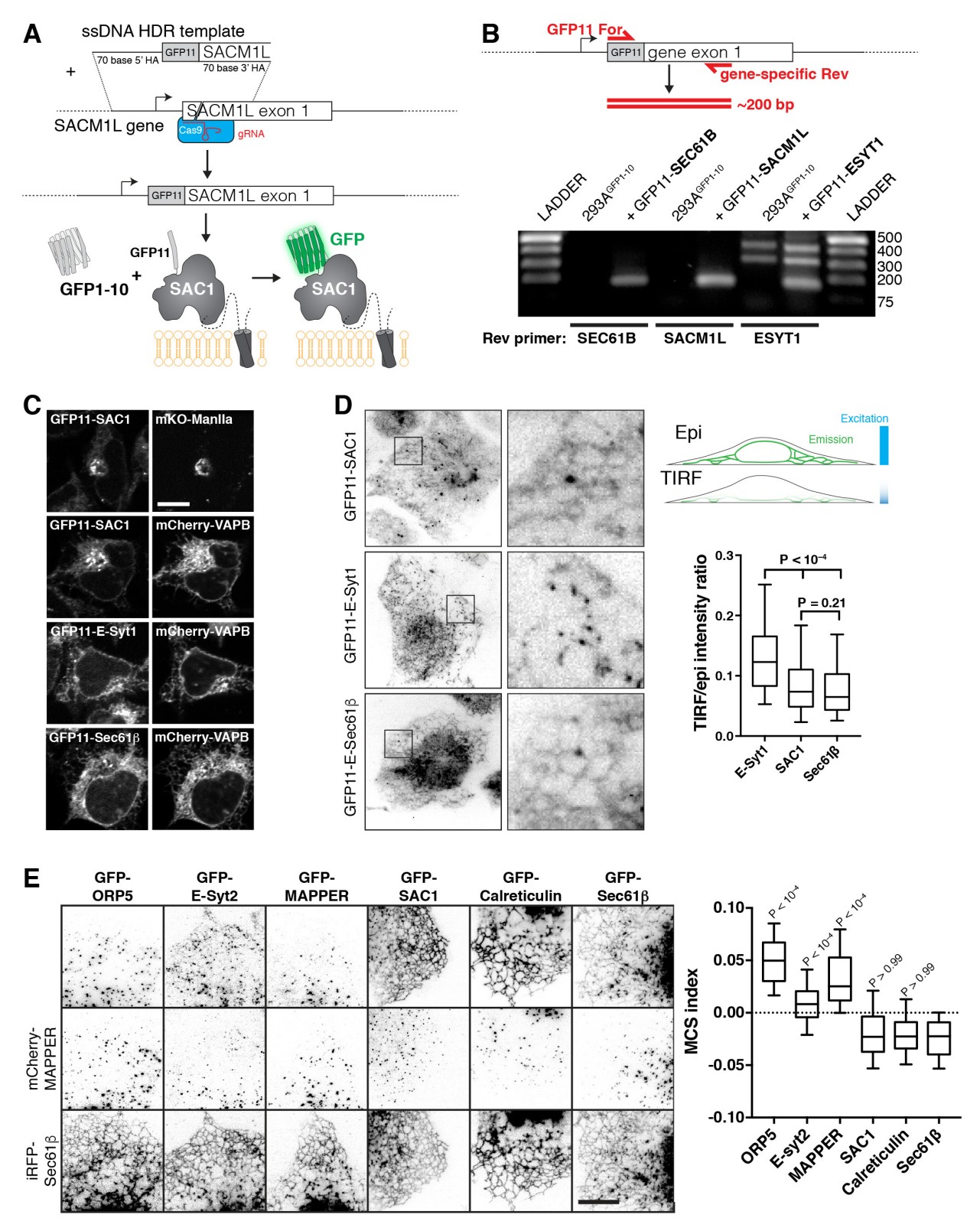

**Figure 2.** Localization of SAC1 relative to ER-PM MCS and ER proteins. (**A**) Strategy for tagging endogenous SAC1: a guide RNA is complexed with Cas9 protein and electroplated into HEK-293A cells with a short single-stranded homology-directed repair (HDR) template. This adds a short tag encoding the 11th strand of the GFP beta barrel. When expressed, this strand assembles with co-expressed GFP1-10 to make functional GFP. (**B**) Specificity of genomic tagging. 293A cells stably over-expressing GFP1-10 and edited with the indicated GFP11 tags were genotyped with GFP11 *Figure 2 continued on next page*

*Figure 2 continued*

specific forward primers and a gene-specific reverse primer located ~200 bp downstream in exon 1. (**C**) Confocal images of GFP11 gene edited cells co-expressing mKo-Manosidase II as a cis/medial Golgi marker, or mCherry-VAPB as an ER marker. (**D**) E-Syt1 shows enrichment at the PM relative to SAC1 and Sec61β. Cells were imaged in both TIRF and epi-illumination, and the fluorescence intensity ratio of the two images was calculated. Boxes represent quartiles, whiskers 5–95 percentile. P values are from Dunn's Multiple Comparisons following a Kruskall-Wallis test (p<10$^{-4}$). Data are from 180 (E-Syt1), 234 (SAC1) or 246 (Sec61β) cells imaged across five independent experiments. Insets = 10 μm. (**E**) Expressed SAC1 is not enriched at ER-PM MCS in COS-7 cells. TIRF images of COS-7 cells transfected for 24 hr with the indicated GFP-tagged plasmid and mCherry-MAPPER to label ER-PM MCS along with iRFP-Sec61β to label total ER. Scale bar = 10 μm. The MCS index is the 'difference of differences' between GFP and iRFP-Sec61β as well as GFP and MAPPER signals. P values are from Dunn's Multiple Comparison test relative to GFP-Sec61β, run as a post-hoc to a Kruskal-Wallis test (p<10$^{-4}$). Box and whiskers are quartiles with 10–90 percentiles of 90 (Sec61β), 92 (Calreticulin), 91 (SAC1), 93 (MAPPER) or 92 (E-Syt2, ORP5) cells imaged across three independent experiments.

DOI: https://doi.org/10.7554/eLife.35588.008

The following source data is available for figure 2:

**Source data 1.** Data for panel 2D.
DOI: https://doi.org/10.7554/eLife.35588.009
**Source data 2.** Data for panel 2E.
DOI: https://doi.org/10.7554/eLife.35588.010

MAPPER to give an 'MCS index' (see Materials and methods). *Figure 2E* shows results from 90 cells across three independent experiments, along with representative images (selected on the basis of values close to the median). As can be seen, GFP-SAC1 localizes to the total ER with a distribution indistinguishable from calreticulin or Sec61β, whereas the contact site proteins all exhibit punctate distributions coinciding with MAPPER, although the association between MAPPER and E-Syt2 is not as tight as it is with ORP5 (*Figure 2E*). Collectively, these data show that, in the resting state, SAC1 does not specifically localize at ER-PM MCS.

Dynamic recruitment of SAC1 to ER-PM MCS has been proposed as a mechanism by which cells can modulate SAC1 'trans' activity and thereby PM inositol lipid abundance (*Dickson et al., 2016*). We therefore used time-lapse TIRF microscopy to follow the dynamic recruitment of ER-resident proteins to MCS during stimulation of cells with phospholipase C (PLC) coupled agonists. PLC induces the hydrolysis of PM PtdIns(4,5)$P_2$ to produce the calcium-mobilizing messenger, Ins$P_3$. The resulting calcium release from ER stores triggers transient recruitment of E-Syt1 to ER-PM MCS (*Giordano et al., 2013*), which may facilitate early re-synthesis of inositol lipid (*Saheki et al., 2016*). Subsequently, calcium release leads to depletion of ER calcium, triggering aggregation of the ER calcium sensor STIM1 and its recruitment to ER-PM MCS to activate store-operated calcium entry across the PM (*Liou et al., 2005*). Finally, the continued breakdown of PtdIns(4,5)$P_2$ yields diacylglycerol, which is converted to phosphatidic acid and exchanged for ER-derived PtdIns by MCS-recruited Nir2, facilitating PtdIns(4,5)$P_2$ replenishment (*Chang et al., 2013*; *Kim et al., 2015*). We expressed GFP-tagged ER and MCS proteins in COS-7 cells for six hours in order to achieve low levels of expression (*Figure 3A*). Stimulation of endogenous PLC-coupled P2Y receptors (*Hughes et al., 2007*) with 100 μM ATP triggered GFP-E-Syt1 to rapidly (peaking within 30 s) and transiently (returning to baseline within 2 min) recruit to punctate ER-PM MCS, whereas STIM1 exhibited a slower (peaking within 2 min) transient recruitment. GFP-Nir2 exhibited a much slower (5 min) recruitment that was sustained during the 10 min experiment. GFP-SAC1, on the other hand, showed no change in its localization close to the PM, nor did the ER-resident proteins GFP-calreticulin or GFP-Sec61β (*Figure 3A*).

We also checked for transient re-localization of SAC1 in our endogenous labelled 293A$^{GFP1-10}$ cells. In this case, stimulation with 100 μM carbachol was used to stimulate PLC via endogenously expressed muscarinic M3 receptors (*Luo et al., 2008*). Carbachol elicited a typical, transient elevation of cytosolic calcium (measured with Fura-red) with a sustained plateau that was unchanged by tagging endogenous proteins with GFP11 (*Figure 3B*). Stimulation caused the rapid, transient recruitment of GFP11-E-Syt1 to puncta (*Figure 3C*), as seen previously with endogenously tagged GFP-E-Syt1 in HeLa cells (*Saheki et al., 2016*). However, no change in the localization of endogenous GFP11-SAC1 or GFP11-Sec61β was observed (*Figure 3C*).

Together, these results show that SAC1 is not specifically enriched at (nor depleted from) ER-PM MCS, even when other proteins are being recruited to these sites to facilitate calcium and inositol

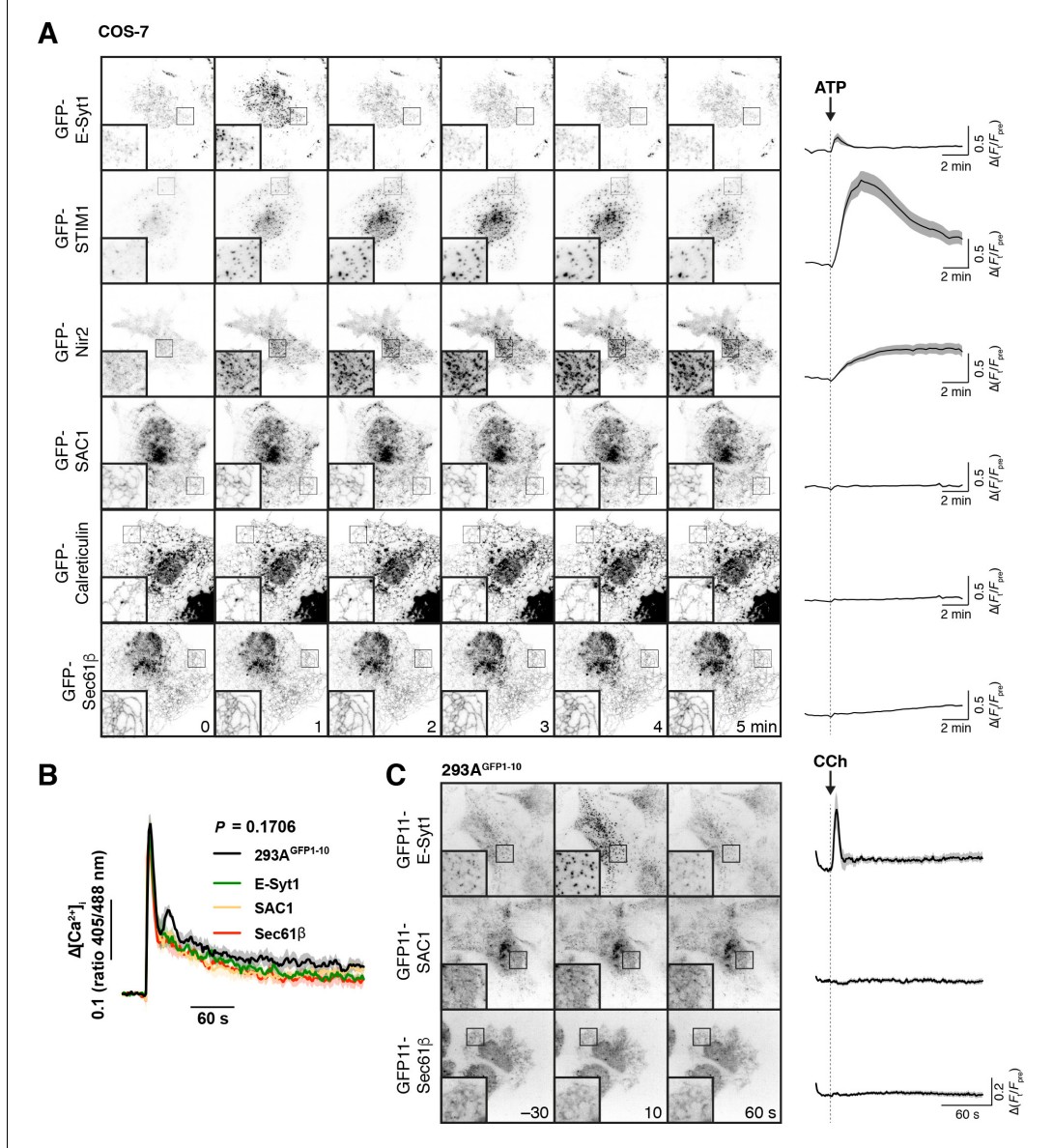

**Figure 3.** Recruitment of proteins to ER-PM MCS. (**A**) Transfected SAC1 does not dynamically re-distribute to ER-PM MCS in COS-7 cells. Time-lapse TIRF microscopy of COS-7 cells transfected with the indicated GFP-tagged proteins for 6–7 hr. Cells were stimulated with 100 μM ATP as indicated. Insets = 10 μm. The traces at right show $\Delta(F_t/F_{pre})$ and are means ± s.e. of 30 (Sec61β, SAC1, Calreticulin), 27 (STIM1), 29 (ESyt1) or 20 (Nir2) cells imaged across three independent experiments. (**B**) Gene edited alleles do not perturb calcium signals. Edited 293A^GFP1-10 cells were loaded with Fura-red and the ratio of fluorescence intensity with respect to 405 and 488 nm excitation was measured. Cells were stimulated with carbachol (CCh) at 30 s to activate phospholipase C signaling. Data are grand means of four experiments (shaded regions represent s.e.). The P value represents results of a two-way ANOVA comparing cell lines. (**C**) Endogenous SAC1 does not recruit to ER-PM contact sites in 293A^GFP1-10 cells. Images show representative gene-edited cells at the indicated times during time-lapse TIRF imaging. Carbachol was added to stimulate phospholipase C signaling at time 0. Images are averages of 5 frames acquired over 10 s to improve signal to noise. Traces represent mean change in fluorescence intensity (normalized to pre-stimulation levels) with s.e. of 40 (E-Syt1), 38 (SAC1) or 37 (Sec61β) cells imaged across five independent experiments.

DOI: https://doi.org/10.7554/eLife.35588.011

The following source data is available for figure 3:

**Source data 1.** Data for panel 3A.
DOI: https://doi.org/10.7554/eLife.35588.012
**Source data 2.** Data for panel 3B.
DOI: https://doi.org/10.7554/eLife.35588.013
**Source data 3.** Data for panel 3C.
DOI: https://doi.org/10.7554/eLife.35588.014

lipid homeostasis. The limited and unchanging localization of SAC1 at MCS therefore appears co-incidental with its well-known distribution throughout the ER.

## 'cis' and 'trans' activity of SAC1 in cells

Although the data presented so far failed to show compelling evidence for 'trans' activity of SAC1, neither could we completely exclude it. Notably, SAC1 was not excluded from ER-PM MCS, so the possibility remained that SAC1 may, given the right circumstances, be able to operate in a 'trans' configuration at MCS. We decided to devise experiments to deduce whether such activity is possible in living cells.

To this end, we designed a strategy utilizing chemically-induced dimerization of FK506-binding protein (FKBP12) and the FKBP and rapamycin binding (FRB) domain of mTOR (*Ho et al., 1996*). PM-anchored FKBP and ER-anchored FRB can be induced to form ectopic ER-PM MCS using this system (*Várnai et al., 2007*), so we reasoned this approach could be used to target SAC1 to ER-PM MCS and assay for 'trans' activity (*Figure 4A*). Likewise, 'cis' activity could be tested simply by replacing the C-terminal TMDs with FKBP (*Figure 4A*), similarly to how the enzyme has been introduced in vitro (*von Filseck et al., 2015b*; *Stefan et al., 2011*; *Mesmin et al., 2013*). A caveat to this approach was that we had already found that expressing SAC1$^{\Delta TMD}$-FKBP leads to reduced PM PtdIns4P, preventing us from measuring further PtdIns4P hydrolysis with our GFP-P4M biosensor (*Figure 1A*). To circumvent this, we turned to a higher-avidity tandem dimer of this domain, GFP-P4M $\times$ 2 (*Hammond et al., 2014*; *Levin et al., 2017*). This reporter was able to detect residual PtdIns4P in the PM of SAC1$^{\Delta TMD}$-FKBP over-expressing COS-7 cells (*Figure 4B*), allowing us to assay for recruitment-induced PtdIns4P depletion.

The results of these experiments are presented in *Figure 4C*. Overall, there were substantial differences in PM PtdIns4P changes reported by GFP-P4M $\times$ 2 depending on the SAC1 chimera used (p<10$^{-4}$, repeated-measures two-way ANOVA). A fusion of FKBP to the N-terminus of SAC1 showed robust recruitment of the enzyme to ER-PM MCS within 1 min of rapamycin addition (see the inset graphs), and we detected a very subtle decline of PM PtdIns4P as compared to the catalytically inactive C389S control, though this did not reach statistical significance (p=0.15, Tukey's multiple comparison test). Fusion of FKBP to the C-terminus of the protein produced a greater depletion of PtdIns4P relative to its C389S control (p=0.007), perhaps because this C-terminal fusion could pull the catalytic domain closer to the PM after complex formation. Therefore, we could induce a very limited 'trans' activity of over-expressed SAC1.

Replacement of the TMD with FKBP permitted SAC1 recruitment to the PM in a 'cis' configuration. This protein expressed poorly in cells, and recruited much less robustly than the 'trans' acting fusions of full-length SAC1; the insets to the graphs in *Figure 4C* show the change in absolute gray levels from identical exposures of the mCherry-tagged GFP-fusions acquired under the same excitation intensity, and therefore represent a relative indication of the mass of protein recruited. An order of magnitude less SAC1$^{\Delta TMD}$-FKBP was recruited than full-length FKBP-SAC1 or SAC1-FKBP. Nevertheless, recruitment was still accomplished in <1 min, and the effect on PM PtdIns4P was much more dramatic than seen with either full-length fusion or the catalytically inactive C389S mutant of this construct (p<10$^{-4}$). Fusion of FKBP to the N-terminus of SAC1$^{\Delta TMD}$ produced better expression and more robust recruitment than the C-terminal fusion, and also a slightly more efficient depletion of PtdIns4P (p=0.0008), demonstrating that fusion of FKBP to the N- or C-termini did not disrupt SAC1 activity. Note that with both $\Delta$TMD fusions, the extent of depletion is most likely an underestimate, since the high avidity P4M $\times$ 2 construct translocated to other PtdIns4P-replete compartments, such as the Golgi and endosomes, after release form the PM (*Levin et al., 2017*). Many of these compartments are visible in the evanescent field of flat COS-7 cells (*Figure 4C*), causing an underestimate in the depletion of PM PtdIns4P, which is measured from total fluorescence intensity in the evanescent field.

It has previously been reported that the ~70 amino acid region between the N-terminal catalytic domain and the TMD of SAC1 are essential for substrate recognition and catalysis (*Cai et al., 2014*). We tested a similar truncation (removing residues 452–587 including these 70 amino acids and the TMD) fused C-terminally to FKBP. Although the protein recruited to the PM robustly, it failed to induce any PtdIns4P depletion (*Figure 4C*), consistent with the prior study.

Collectively, these results demonstrate that SAC1 has very robust activity when it meets its substrate in a 'cis' configuration in the cellular environment, with much less activity in 'trans'. We

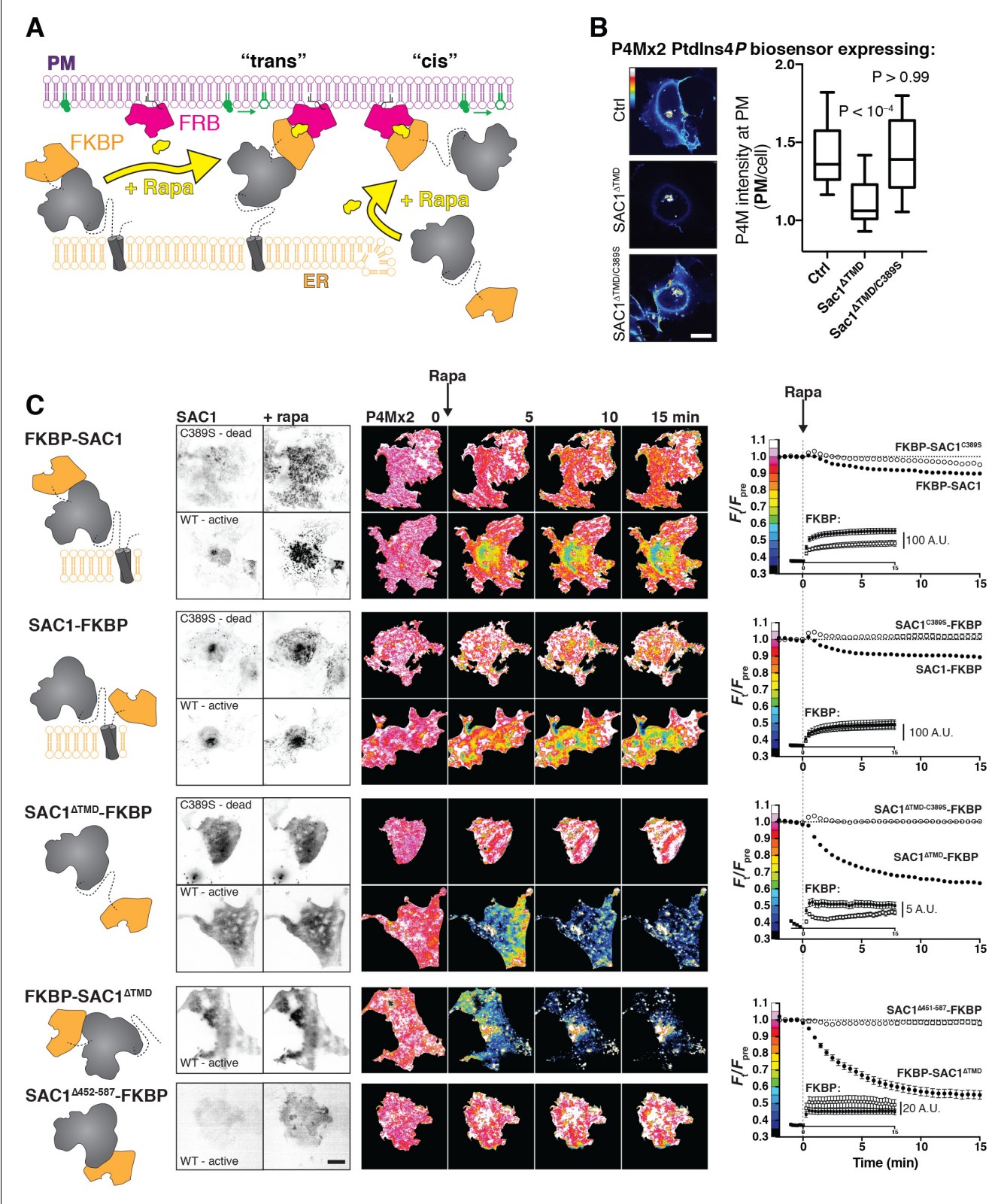

**Figure 4.** SAC1 is much more active at the PM in 'cis'. (**A**) Strategy to recruit SAC1 to the PM in 'cis' or 'trans' using the FRB/FKBP12 heterodimerization system. (**B**) PM PtdIns4*P* is still detectable at the PM with P4M × 2 after transfection with SAC1^ΔTMD. COS-7 cells transfected with GFP-P4M and either FKBP-mCherry (Ctrl), SAC1^ΔTMD-FKBP-mCherry or catalytically inactive SAC1^ΔTMD/C389S were imaged live by confocal microscopy. Representative images are shown (bar = 20 μm). The graph shows P4M intensity at the plasma membrane (defined by CellMask deep red dye) normalized to total cell

*Figure 4 continued on next page*

Figure 4 continued

intensity; box and whisker plot shows quartiles and 5–95 percentiles of 90 cells from three independent experiments. P values derive from Dunn's multiple comparison test compared to Ctrl after a Kruskal-Wallis test ($p < 10^{-4}$). (**C**) Recruitment of SAC1 to the PM in 'cis' is far more effective in depleting PtdIns4$P$ than it is in 'trans'. TIRF images of COS-7 cells transfected with a Lyn$_{11}$-FRB-iRFP PM recruiter, the indicated mCherry-tagged SAC1-FKBP or FKBP-SAC1, and GFP-P4M × 2. Graphs show means ± s.e. Images are representative of n cells, x independent experiments: 57, 6 (FKBP-SAC1); 41, 4 (FKBP-SAC1$^{C389S}$); 28, 3 (SAC1-FKBP); 30, 3 (SAC1-FKBP); 57, 6 (SAC1-FKBP); 36, 4 (SAC1-FKBP); 26, 3 (FKBP-SAC1); 29, 3 (SAC1$^{\Delta452-587}$). Inset graphs show the raw change in signal intensity for the mCherry-FKBP tagged SAC1 chimeras. Images of GFP-P4M × 2 are normalized to the mean pre-stimulation pixel intensity, that is $F_t/F_{pre}$ with the color coding reflected in the graph y-axis. Scale bar = 20 μm.

DOI: https://doi.org/10.7554/eLife.35588.015

The following source data is available for figure 4:

**Source data 1.** Data for panel 4B.
DOI: https://doi.org/10.7554/eLife.35588.016
**Source data 2.** Data for panel 4C.
DOI: https://doi.org/10.7554/eLife.35588.017
**Source data 3.** Data for panel 4C (insets).
DOI: https://doi.org/10.7554/eLife.35588.018

hypothesized that this could reflect insufficient length of the cytosolic N-terminal region to orient the catalytic domain for efficient 'trans' catalysis. We further hypothesized that increasing the length between the TMD and the cytosolic domain could overcome this deficit (***Figure 5A***). To this end, we generated chimeras of our FKBP-SAC1 constructs containing repeats of the helical linker sequence EAAAR (***Yan et al., 2007***). Each repeat of this sequence, when forming an α-helix, should have a length of approximately 7.5 Å; our tandem repeats of (EAAAR)$_{2-10}$ (termed HLx2-10) therefore have predicted lengths of 1.5–7.5 nm.

As shown in ***Figure 5B and C***, all of these helical linker SAC1 chimeras were efficiently recruited to ER-PM MCS within 1 min of the addition of rapamycin. Insertion of 2–6 repeats did not produce a significant enhancement of PM PtdIns4$P$ hydrolysis (***Figure 5B and D***; p≥0.44, 57 WT or 30 HLxN cells, Tukey's multiple comparisons test after a repeated measure 2-way ANOVA, $p < 10^{-4}$). However, insertion of 8 helical repeats produced substantial depletion of PM PtdIns4$P$ after recruitment to ER-PM MCS ($p < 10^{-4}$), which was not enhanced by the addition of a further two helical repeats to produce HLx10 (p=0.96, 30 cells each). No depletion of PtdIns4$P$ was observed with a catalytically inactive C389S mutant SAC1 containing the HLx8 linker (p=0.96, 30 cells). Clearly, the addition of between 6 and 7.5 nm of additional linker between the catalytic domain and the TMD endows a robust 'trans' catalytic activity on SAC1.

These experiments demonstrated a weak capacity of SAC1 chimeras to act in 'trans' when they were forced into a conformation spanning the ER-PM junction by heterodimerization. This is likely a poor representation of the physiologic state, wherein other proteins would mediate ER-PM MCS tethering and SAC1 would not be subject to such conformational constraints. To test for the propensity of SAC1 to act in 'trans' at ER-PM MCS under unconstrained conditions, we capitalized on the observation that extended synaptotagmins expand MCS when over-expressed (***Giordano et al., 2013***; ***Fernández-Busnadiego et al., 2015***; ***Figure 6A***). These expanded contact sites would be expected to present more endogenous 'trans' acting SAC1 to the plasma membrane, thus depleting PM PtdIns4$P$ (***Dickson et al., 2016***). However, we found that expanded ER-PM MCS induced by E-Syt2 over-expression had no effect on PtdIns4$P$ biosensor localization at the PM (***Figure 6B***). In contrast, extensive depletion was observed after over-expression of ORP5, which is known to deplete PtdIns4$P$ by transferring it back to the ER (***Chung et al., 2015***; ***Sohn et al., 2016***).

For a more acute interrogation of 'trans' activity, we next induced dimerization between PM targeted FRB and an ER-localized FKBP (fused to the membrane anchor of cytochrome B5A; ***Komatsu et al., 2010***) to generate new ER-PM MCS (***Figure 6A***). These induced contact sites formed in under a minute at existing ER-PM MCS marked by GFP-E-Syt2 or GFP-ORP5 (***Figure 6C***). In fact, as soon as they formed, the induced contact sites forced E-Syt2 and ORP5 to the periphery, indicating that the newly formed contacts were too narrow to accommodate native contact site proteins (***Figure 6C***). This 'squeezing out' of the contact site markers was dependent on ER-PM bridging, since GFP-Sec61β, which is not constrained to the PM-facing side of the ER, was not pushed to the periphery (***Figure 6C***).

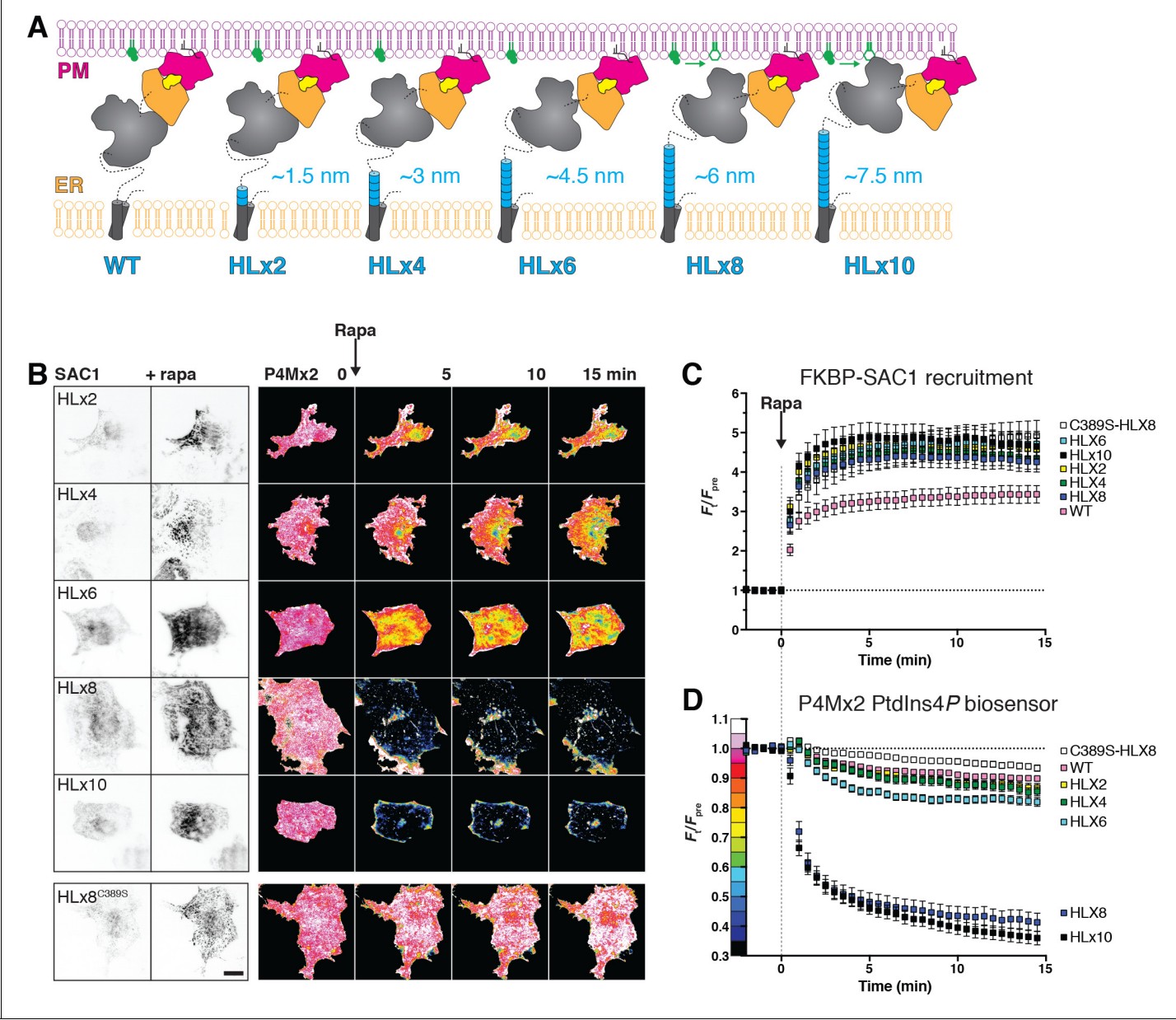

**Figure 5.** An extended helical linker confers 'trans' activity to SAC1. (**A**) Helical linkers (HL) added to FKBP-SAC1 at the end of the first transmembrane domain. Each helical repeat consists of the amino acids EAAAR, expected to form a helix approximately 0.75 nm long. (**B**) TIRF imaging of PtdIns4*P* before and after direct recruitment of SAC1 to ER-PM MCS. TIRF images of COS-7 cells transfected with a Lyn$_{11}$-FRB-iRFP PM recruiter, the indicated mCherry-tagged SAC1-FKBP and GFP-P4M × 2. Images are representative of 30 cells from three independent experiments. Images of GFP-P4M × 2 are normalized to the mean pre-stimulation pixel intensity, that is F$_t$/F$_{pre}$ with the color coding reflected in the graph y-axis of D. Scale bar = 20 µm. (**C**) Helical linkers do not impair recruitment efficiency of FKBP-SAC1. (**D**) FKBP-SAC1-HLx8 and -HLx10 have 'trans' activity. Graphs in C and D show fluorescence intensity in the TIRF footprint of each cell for mCherry-tagged FKBP-SAC1 or GFP-tagged P4M × 2, respectively. Data are means ± s.e., 30 cells for all except WT, with 57 cells. Data for the wild-type FKBP-SAC1 is re-plotted from *Figure 4*.

DOI: https://doi.org/10.7554/eLife.35588.019

The following source data is available for figure 5:

**Source data 1.** Data for panel 5C.
DOI: https://doi.org/10.7554/eLife.35588.020
**Source data 2.** Data for panel 5D.
DOI: https://doi.org/10.7554/eLife.35588.021

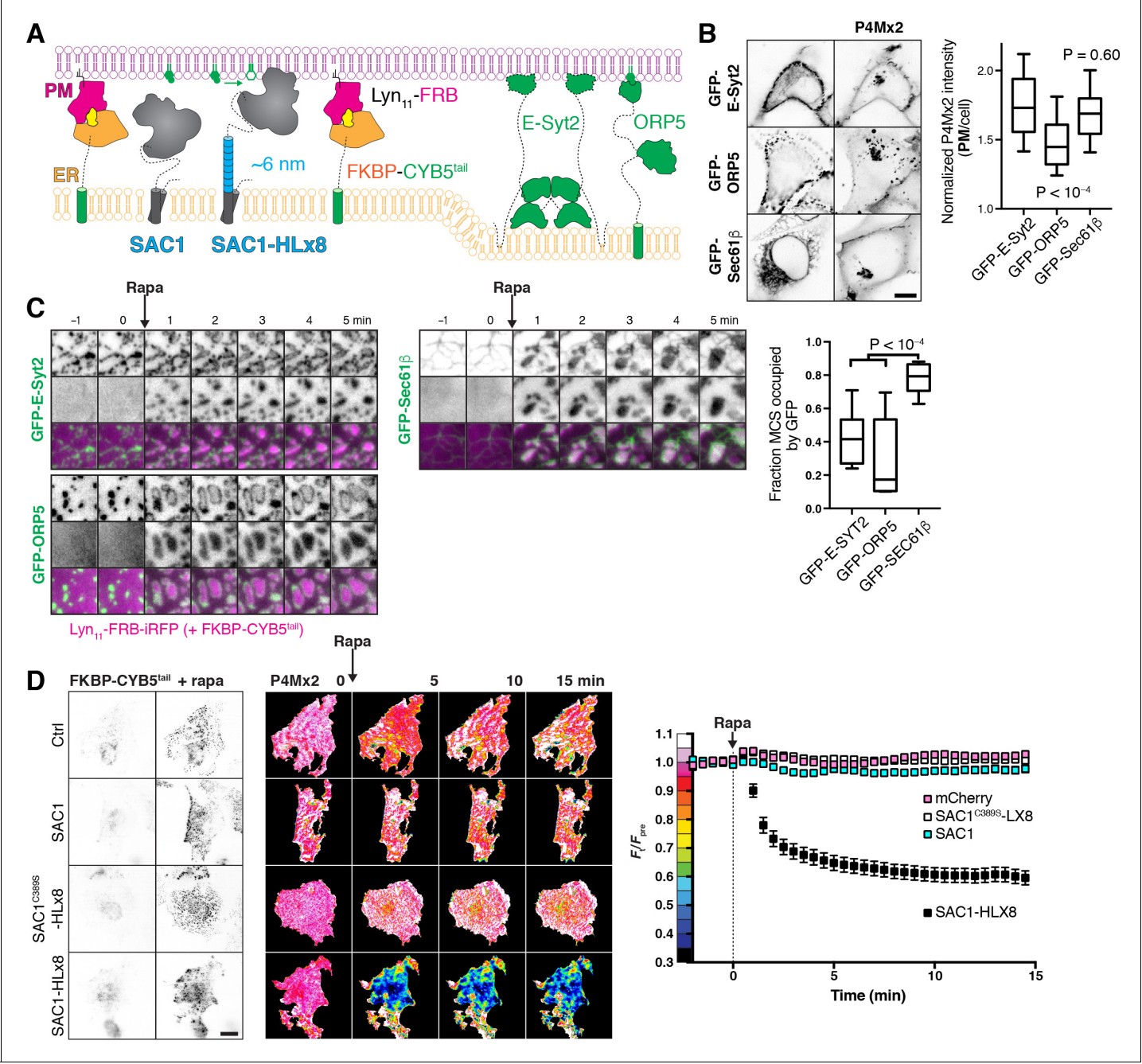

**Figure 6.** An extended helical linker is required for 'trans' activity of SAC1 at induced ER-PM MCS. (**A**) Induction of artificial ER-PM MCS using rapamycin-induced dimerization of PM Lyn$_{11}$-FRB and ER FKBP-CYB5A$^{tail}$. (**B**) Over-expression of E-Syt2 does not deplete PtdIns4$P$. COS-7 cells over-expressing GFP-tagged E-Syt2, ORP5 along with mCherry-P4M × 2; scale bar = 10 µm. Graph shows P4M intensity at the plasma membrane (defined by CellMask deep red dye) normalized to total cell intensity; box and whisker plot shows quartiles and 5–95 percentiles of 89–90 cells from three independent experiments. P values derive from Dunn's multiple comparison test compared to Ctrl after a Kruskal-Wallis test (p<10$^{-4}$). (**C**) FKBP-CYB5$^{tail}$ induces narrower contact sites than those occupied by E-Syt2 or ORP5. COS-7 cells expressing the indicated GFP-fusion protein, Lyn$_{11}$-FRB-iRFP or mCherry-FKBP-CYB5$^{tail}$ (not shown), dimerization induced with Rapa as indicated. Graph shows the fraction of induced contact sites occupied by GFP-fluorescence after 5 min of rapa treatment; box and whisker plot shows quartiles and 5–95 percentiles of 14–19 cells from four independent experiments. P values derive from Dunn's multiple comparison test compared to Ctrl after a Kruskal-Wallis test (p<10$^{-4}$). (**D**) An extended helical linker is required for robust 'trans' activity of SAC1 at ER-PM MCS. Images of TagBFP2-tagged FKBP-CYB5 and GFP-P4M × 2 in COS-7 cells co-transfected with iRFP-tagged Lyn$_{11}$-FRB and the indicated mCherry-tagged SAC1 construct, or mCherry alone as control. Images of GFP-P4M × 2 are normalized to the mean pre-stimulation pixel intensity, that is F$_t$/F$_{pre}$ with the color coding reflected in the graph y-axis. Scale bar = 20 µm. Graphs show the

*Figure 6 continued on next page*

*Figure 6 continued*

fluorescence intensity of GFP-P4M × 2 in the TIRF footprint of each cell (means ± s.e., 29–30 cells from three independent experiments) normalized to the mean pre-stimulation level ($F_{pre}$).

DOI: https://doi.org/10.7554/eLife.35588.022

The following source data is available for figure 6:

**Source data 1.** Data for panel 6B.
DOI: https://doi.org/10.7554/eLife.35588.023
**Source data 2.** Data for panel 6C.
DOI: https://doi.org/10.7554/eLife.35588.024
**Source data 3.** Data for panel 6D.
DOI: https://doi.org/10.7554/eLife.35588.025

We next tested PM PtdIns4$P$ abundance after inducing these ER-PM MCS in cells expressing either endogenous SAC1 (and mCherry as a control), or over-expressing mCherry-tagged SAC1, SAC1-HLx8 or SAC1$^{C389S}$-HLx8 (*Figure 6A*). Addition of rapamycin produced robust formation of ER-PM MCS in all cases (*Figure 6D*), but with strikingly different results on PM PtdIns4$P$ (*Figure 6D*): co-expression of WT SAC1, SAC1$^{C389S}$-HLx8 or mCherry alone did not produce a noticeable change in PM PtdIns4$P$, and were not significantly different from one another (p≥0.51 Tukey's multiple comparisons test, 29–30 cells after a repeated measures two-way ANOVA comparing proteins, $p<10^{-4}$); whereas SAC1-HLx8 produced a robust and rapid depletion of PtdIns4$P$ relative to the other proteins ($p<10^{-4}$, 30 cells) after induction of ER-PM MCS. We interpret these data to mean that SAC1 has robust 'trans' activity only when an extended helical linker is introduced between the TMD and catalytic regions; it follows that the native enzyme is unable to produce a conformation that orients the catalytic domain for efficient 'trans' catalysis at ER-PM MCS in vivo.

## Discussion

We undertook this study to answer the simple question: does the ER-resident SAC1 lipid phosphatase act in 'cis' or 'trans' in the cellular context? This question is important, since a 'cis' acting SAC1 spatially segregates PtdIns4$P$ metabolism, allowing it to drive a PPInMF, whilst a 'trans' acting SAC1 could explain the control of PtdIns4$P$ biosynthesis that regulates other metabolic and trafficking functions of the lipid. We demonstrated that: (i) agents that acutely inhibit SAC1 activity in cells produce the accumulation of PtdIns4$P$ substrate in the ER; (ii) under either resting or MCS forming conditions, SAC1 fails to enrich at ER-PM MCS, the would-be sites of 'trans' activity; (iii) SAC1 has robust 'cis' activity but very poor 'trans' activity in living cells; (iv) for appreciable 'trans' activity, an additional ~6 nm linker between the ER localized TMD and the catalytic domain must be introduced. Collectively, these data demonstrate that the enzyme's mode of activity inside a living, mammalian cell is in 'cis'.

How can these data be reconciled with previous assertions of 'trans' activity? In vitro reconstitution approaches have yielded convincing evidence for both 'cis' and 'trans' activity (*Stefan et al., 2011*; *Mesmin et al., 2013*). However, these approaches are unlikely to faithfully recapitulate the cellular environment, with all of its constraints inherent to the complex molecular milieu. Therefore, a conclusive picture will only emerge from studies of intact cellular systems. 'Trans' activity of SAC1 was previously reported to be supported in yeast by the protein Osh3p at ER-PM MCS (*Stefan et al., 2011*), though this protein is now believed to facilitate PtdIns4$P$ transfer, and so these data can equally be interpreted as supporting 'cis' activity of SAC1 (*Wong et al., 2017*). Similarly, dynamic recruitment of SAC1 to ER-PM MCS was recently proposed to modulate PM inositol lipid synthesis as a feedback mechanism (*Dickson et al., 2016*): it was shown that dynamic relocation of SAC1 from ER-PM MCS during PLC activation functioned as a rheostat, reducing PtdIns4$P$ catabolism and hence supporting PtdIns(4,5)$P_2$ re-synthesis. Although a 'trans' activity of SAC1 was suggested to be at work, we suggest that a 'cis' acting SAC1 is equally capable of supporting such a mechanism: notably, PLC activity after activation of highly expressed muscarinic receptors reduces both PM PtdIns(4,5)$P_2$ and PtdIns4$P$ (*Horowitz et al., 2005*), which will disrupt attachment of the ER-PM tethering E-Syt2 protein (*Giordano et al., 2013*) as well as the PtdIns4$P$-transporting ORP5 and ORP8 proteins (*Chung et al., 2015*). Therefore, the dynamic relocation of SAC1 from ER-PM

MCS is most likely coincidental to the loss of ER-PM MCS proteins that support PtdIns4$P$ transport to a 'cis' acting ER SAC1.

Over-expression of full-length SAC1 was shown to both reduce steady-state PtdIns4$P$ levels at the PM and Golgi (*Hammond et al., 2014*) and accelerate its rate of degradation from the PM (*Sohn et al., 2016*). These data are most easily interpreted as being due to 'trans' activity. However, we believe these observations can be reconciled with a 'cis' acting SAC1. The yeast homologues of ORP5 and ORP8 that facilitate PM PtdIns4$P$ transport, Osh6p and Osh7p, have a higher affinity for their inositol lipid ligand than for their cargo lipid, phosphatidylserine (*Moser von Filseck et al., 2015a*). Given that in mammalian cells, most of this cargo lipid is present in the lumenal leaflet of the ER (*Fairn et al., 2011*), its abundance will be low in the cytosolic leaflet. Since we now show that SAC1 appears not to be enriched at ER-PM MCS, it therefore seems likely that ORP5 and 8 (or Osh6p and Osh7p) may be able to back-traffic PtdIns4$P$ before it is degraded by SAC1; that is, futile cycles of PtdIns4$P$ transfer from PM to ER and back can occur. SAC1 over-expression would therefore be expected to decrease such futile reactions and thus deplete PM PtdIns4$P$. A similar mechanism could be at work at other PtdIns4$P$-replete membranes.

Our data clearly demonstrate that SAC1 has 'cis' activity in mammalian cells. However, we also showed that the only way to stimulate 'trans' activity was to either artificially tether the protein to contact sites, which resulted in poor activity (*Figure 4*), or to extend the linker between TMD and catalytic domain by ~6 nm (*Figure 5*, *Figure 6*), yielding more robust activity. SAC1's 'reach' in its native form was recently proposed to be no more than ~7 nm (*Gatta et al., 2017*), whereas the dimensions of native ER-PM MCS vary from 15 to 25 nm in COS-7, depending on the tethers involved (*Fernández-Busnadiego et al., 2015*). We measured SAC1 recruitment at ER-PM MCS that were artificially induced and appear to have dimensions significantly shorter than this, yet the enzyme still did not produce robust 'trans' activity. It therefore seems clear that SAC1 would be unable to exhibit 'trans' activity at native ER-PM MCS.

What are the implications for an obligatory 'cis' acting SAC1? First and foremost, it meets a strict requirement for the enzyme's role in generating a PPInMF that can drive non-vesicular lipid transport (*Mesmin et al., 2013*). A SAC1 unable to dephosphorylate PtdIns4$P$ in 'trans' at a MCS, but able to efficiently degrade ER PtdIns4$P$ in 'cis' ensures the maintenance of a steep concentration gradient of the lipid between target membrane and the ER. The potential energy of this gradient can then be efficiently harnessed to drive vectorial transfer of other lipid cargos. In short, 1 mol of ATP (per mol PtdIns4$P$ synthesized) is expended for each mol of lipid transferred with a 'cis' acting SAC1, whereas unconstrained 'trans' SAC1 activity would force the ratio of ATP expenditure to lipid transfer to increase.

Another important implication of an obligate 'cis' acting SAC1 is how the enzyme can participate in PtdIns4$P$ homeostasis and function more generally. To fully appreciate the implications, it is necessary to consider the other SAC domain containing proteins executing PtdIns4$P$ catabolism. These included SAC2, Synaptojanin (Synj) one and Synj2 (*Chung et al., 1997*; *Hsu et al., 2015*; *Nakatsu et al., 2015*; *Khvotchev and Südhof, 1998*). SAC2 has recently been identified as operating on early and recycling endosomes (*Hsu et al., 2015*; *Nakatsu et al., 2015*), whereas Synj isoforms appear to participate in clathrin-mediated endocytosis (*Perera et al., 2006*; *Rusk et al., 2003*). Together with a 'cis' acting SAC1, it thus appears that PM PtdIns4$P$ degradation occurs only when the lipid leaves the PM via vesicular (SAC2, Synjs) or non-vesicular traffic (SAC1). In effect, traffic is a key regulator of PM PtdIns4$P$ metabolism.

A 'cis' acting SAC1 is easier to reconcile with respect to Golgi PtdIns4$P$ metabolism. A PPInMF acting at the (TGN) has been implicated in sterol traffic from the ER into the secretory pathway (*Mesmin et al., 2013*; *von Filseck et al., 2015b*), necessitating a lack of SAC1 activity in the TGN. However, PtdIns4$P$'s selective enrichment at the TGN has been shown to be controlled by the traffic of SAC1 to cis/medial Golgi membranes (*Cheong et al., 2010*), where it is activated by Vps74/GOLPH3 (*Cai et al., 2014*; *Wood et al., 2012*). A lack of proliferative signals stimulates traffic of SAC1 from ER to Golgi, causing reduction of TGN PtdIns4$P$ pools and a block of secretory vesicular traffic (*Blagoveshchenskaya et al., 2008*). Under these conditions, the resulting ablation of the TGN-associated PPInMF and shut down of sterol egress would make sense.

Turnover of LEL PtdIns4$P$ may also be dependent on PtdIns4$P$ transfer to ER-associated SAC1, since SAC2 and the Synj isoforms are not poised to operate at this compartment. Recently, the PtdIns4$P$ transfer activity of LEL-localized ORP1L was shown to be critical for sterol transfer

(*Zhao and Ridgway, 2017*); however, operation of a LEL PPInMF does not make sense in this context, since the sterol concentration in lysosomal membranes is expected to be high and flux into the ER would be down a concentration gradient (*Zhao and Ridgway, 2017*); lysosomal PtdIns4P would actually compete with sterol efflux. Instead, a reciprocal relationship between LEL sterol content and PtdIns4P synthesis is expected, which is not consistent with the activation of LEL-associated PI4KIIα by sterols (*Waugh et al., 2006*). Clearly, the relationship between LEL PtdIns4P turnover and lipid traffic is incompletely understood at present.

To conclude, we present evidence that SAC1 is an obligate 'cis' acting enzyme, competent to degrade its PtdIns4P substrate only in its resident ER and Golgi membranes. When considered in the context of MCS-localized OSBP-related PtdIns4P transfer proteins, this 'cis'-acting SAC1 can account for all of the currently assigned functions for this enzyme, including driving a PPInMF between recipient organelles and the ER, as well as controlling PtdIns4P abundance in different organelles. It does, however, implicate a role for non-vesicular transport of inositol lipids in their homeostasis and regulation of their downstream functions.

## Materials and methods

### Cell culture and transfection

COS-7 African Green monkey fibroblasts were obtained from ATCC (CRL-1651; RRID: CVCL_0224) and 293A cells (a HEK 293 subclone with flat morphology; R70507; RRID:CVCL_6910) were obtained from ThermoFisher. Cell lines were handled independently to prevent cross-contamination, and were screened regularly for mycoplasma contamination with Hoechst staining. Cell lines were propagated to no more than passage 30. They were cultivated in growth medium consisting of DMEM (low glucose, glutamax supplement, pyruvate; ThermoFisher 10567022) supplemented with 10% heat-inactivated fetal bovine serum (ThermoFisher 10438–034), 100 u/ml penicillin, 100 µg/ml streptomycin (ThermoFisher 15140122) and 0.1% chemically-defined lipid supplement (ThermoFisher 11905031) in 75 cm$^2$ vented tissue culture flasks. Twice per week, almost confluent cultures were rinsed in PBS and suspended with 1 ml TrpLE Express (no phenol red; ThermoFisher 12604039) and diluted 1:5 for propagation in fresh flasks. For experiments, cells were seeded at 12.5–50% confluence on 10 µg/ml fibronectin (ThermoFisher 33016–015)-coated 35 mm dishes containing 20 mm #1.5 glass bottoms (CellVis D35-22-1.5-N) in 2 ml growth medium.

Cells were transfected 1–24 hr after seeding, once they had reached 25% (TIRF) or 50% (confocal) confluence. 0.5–1 µg plasmid DNA was complexed with 3 µg Lipofectamine 2000 (ThermoFisher 11668019) in 200 µl Opti-MEM (ThermoFisher 51985091) for >5 min before adding to the cells. Cells were then used for experiments 6 or 18–24 hr post transfection.

### Reagents

Rapamycin (Fisher Scientific BP2963-1) was dissolved in DMSO to 1 mM. ATP (Sigma 10127523001) was dissolved to 100 mM in 200 mM Tris base with 100 mM MgCl$_2$. Carbachol (Fisher Scientific AC10824-0050) was dissolved in water to 50 mM. bpV(HOpic) (EMD Millipore 203701) was dissolved in DMSO at 10 mM. All were stored as aliquots at –20 ˚C. Hydrogen peroxide (30% solution; EMD Millipore HX0635-3) was stored at 4 ˚C and diluted fresh before use. Fura-red -AM (ThermoFisher F3021) was dissolved in 20% pluronic F-127 (ThermoFisher P3000MP) to 1 mg/ml before use and stored at –20 ˚C. CellMask deep red (ThermoFisher C10046) was stored at –20 ˚C and thawed before dilution.

### Plasmids

Plasmids were constructed in the former Clontech pEGFP-C1 and -N1 backbones. For the most part, the following fluorescent protein fusions were utilized: Unless stated, EGFP refers to *Aequorea victoria* GFP with F64L and S65T mutations (*Cormack et al., 1996*) with human codon optimization. mCherry is an optimized *Discoma* DsRed monomeric variant (*Shaner et al., 2004*). iRFP is the iRFP713 variant of *Rhodopseudomonas palustris* bacteriophytochrome BphP2-derived near-infrared fluorescent protein (*Shcherbakova and Verkhusha, 2013*). mTagBFP2 is an optimized blue-fluorescing mutant of the sea anemone *Entacmaea quadricolor* GFP-like protein eqFP578 (*Subach et al., 2011*).

Plasmids in *Table 1* were constructed using NEB HiFi assembly (New England Biolabs E5520S) or standard restriction cloning.

All plasmids were verified by Sanger sequencing; plasmids generated in this study are available from Addgene (www.addgene.org). Note, the *SACM1L* gene used in this study and previous publications (*Sohn et al., 2016*) contains a missense mutation Y433F relative to the human genome reference sequence. However, the short Genetic Variations database (dbSNP) shows that this allele (rs1468542) represents approximately 60% of alleles present in the human population sampled to date. This allele can therefore be viewed as 'wild-type'.

## Generation on 293A$^{GFP1-10}$ cell line

To create a cell line stably expressing GFP-1–10 for complementation with GFP-11 tags, the Piggy-Bac Transposon system was applied via transfection in 293A cells. Cells were seeded onto 6-well plates and plasmid containing the GFP-1–10 sequence under a CAG promoter and flanked by the proper inverted terminal repeats (APX1-GFP-1–10) was transfected along with plasmid coding the PiggyBac Transposase (CDV-hyPBase) (0.7 µg and 0.3 µg, respectively) as described above. Following overnight transfection, media was replaced with fresh growth media and cells were propagated for 1 week to allow dilution of any free plasmid. Eight independent samples were split in limiting 1:2 dilutions across 12 columns of a 96-well plate. After growth, populations were chosen from the last four columns and a sample of each was screened using an mCherry-SACM1L-GFP-11 reporter plasmid to observe GFP-complementation and assess the efficiency of GFP-1–10 insertion. The polyclonal population of a single well that showed >90% GFP complementation of visibly transfected cells was chosen and propagated for use in future gene-editing experiments (named here as 293A$^{GFP-1-10}$).

## Generation of endogenously tagged cell lines

Single-guide RNA (sgRNA) and homologous-directed repair (HDR) template design followed the method described by (*Leonetti et al., 2016*), utilizing the published sequences available for targeting *SEC61B* and *SACM1L. ESYT1* sgRNA design was informed by the published sequence used by (*Saheki et al., 2016*). All HDR templates included 70 bp homology-arms and the following GFP-11 and flexible linker sequence for genomic insertion (CGTGACCACATGGTCCTTCATGAGTATGTAAA TGCTGCTGGGATTACAGGTGGCGGC). Single-stranded HDR templates were ordered from IDT as ultramers. ssDNA primers to serve as templates for sgRNA production were ordered from Thermo-Fisher. Templates were made by annealing published primers ML611, T25 and BS7 with each site-specific primer following the procedure of Leonetti et al. Annealed products were column purified using the GeneJet Gel Extraction and DNA Cleanup Micro Kit (Thermo Scientific, #K0832). The GeneArt Precision gRNA Synthesis Kit (Thermo Fisher Scientific, A29377) was used for in vitro transcription of DNA templates to produce column-purified sgRNAs. Purity was checked by agarose gel electrophoresis. Cas9 Ribonucleoprotein (RNP) formation and delivery followed the procedure outlined by New England BioLabs (https://www.neb.com/protocols/2016/07/26/electroporation-of-cas9-rnp-ribonucleoprotein-into-adherent-cells-using-the-neon-electroporation). 2.1 µL of GeneArt Platinum Cas9 Nuclease (ThermoFisher, B25640) was incubated with 0.5 µL sgRNA (~10 µM) in Buffer R of the Neon Electroporation System (ThermoFisher, MPK1025) at room temperature for 20 min to form Cas9 RNP complexes. Meanwhile 293A$^{GFP-1-10}$ cells were prepared from 90% confluent T75 culture flasks to obtain $1-2 \times 10^6$ cells suspended in 50 µL Buffer R. 2 µL of HDR template (100 mM) was then added to the incubating RNPs. 5 µL of the prepared cells were then added to the RNP incubation tubes, and 10 µL of this mixture was aspirated with the Neon pipette and electroporated (1500 V, 20 ms, one pulse). Contents of the Neon tip were then immediately transferred to a single well of a 6-well plate containing 2 mL of pre-warmed antibiotic-free complete DMEM and allowed to recover. After recovery and growth, electroporated cells were screened with confocal microscopy. Populations containing correctly edited cells were enriched by fluorescence-activated cells sorting using a Rheum Aria sorter (University of Pittsburgh Flow Cytometry Core). After sorting, genomic DNA was isolated using the PureLink Genomic DNA Mini Kit (ThermoFisher K1820-01). A standard GFP-11 forward primer and gene-specific reverse primers were used to compare the presence of ~200 bp amplicons from each edited cell line to the non-edited 293A$^{GFP-1-10}$ cells. Additionally, using gene-specific primers ~ 75 bp upstream of the GFP-11 insertions, amplicons were

**Table 1.** Plasmids used in this study.
Genes are human unless otherwise stated

| Plasmid | Backbone | Insert | Ref |
|---|---|---|---|
| APX1-GFP1-10 | APX1 | super-folder GFP | This study |
| CDV-hyPBase | pigg | *piggyBAC* transposase | (*Yusa et al., 2011*) |
| NES-EGFP-P4M × 1 | pEGFP-C1 | *X.leavis map2k1.L*(32-44):EGFP:*L. pneumophila SidM*(546-647) | This study |
| FKBP-mCherry | pmCherry-N1 | *FKBP1A*(isoform a, 3–108):mCherry | This study |
| SAC1$^{\Delta TMD}$-FKBP-mCherry | pmCherry-N1 | *SACM1L*(1-521):*FKBP1A*(3-108):mCherry | This study |
| SAC1$^{C389S\Delta TMD}$-FKBP-mCherry | pmCherry-N1 | *SACM1L*(C389S; 1–521):*FKBP1A*(3-108):mCherry | This study |
| iRFP-Sec61β | piRFP-C1 | iRFP:*SEC61B* | This study |
| mKO-ManII | pmKO-N1 | Kusabira Orange 2:*Man2a*(1-102) | Tamas Balla |
| mCherry-VAPB | pmCherry-C1 | mCherry:*VAPB* | This study |
| mCherry-MAPPER | pmCherry-C1 | mCherry:MAPPER | (*Chang et al., 2013*) |
| EGFP-MAPPER | pEGFP-C1 | EGFP:MAPPER | (*Chang et al., 2013*) |
| GFP-ORP5 | pEGFP-C1 | EGFP:*OSBPL5*(isoform a) | (*Sohn et al., 2016*) |
| GFP-E-Syt2 | pEGFP-C1 | EGFP:*ESYT2* | (*Giordano et al., 2013*) Addgene plasmid #66831 |
| EGFP-SAC1 | pEGFP-C1 | EGFP:*SACM1L* | (*Sohn et al., 2016*) |
| mEmerald-N16-Calreticulin | pmEmerald-N1 | mEmerald:*CALR* | Michael Davidson (Addgene plasmid #54023) |
| GFP-Sec61β | pAcGFP-C1-Sec61β | *Aequorea coerulescens* GFP:*SEC61B* | (*Voeltz et al., 2006*) Addgene plasmid #15108 |
| EGFP-E-Syt1 | pEGFP-C1 | EGFP-*ESYT1* | (*Giordano et al., 2013*) Addgene plasmid #66830 |
| EGFP-STIM1 | pEGFP-C1 | *STIM1*(isoform 1 1–22):EGFP:*STIM1*(23–791) | (*Várnai et al., 2007*) |
| EGFP-Nir2 | pEGFP-N1 | EGFP:*PITPNM1*(isoform 2) | (*Kim et al., 2015*) |
| EGFP-P4M × 2 | pEGFP-C1 | EGFP:*L. pneumophila SidM*(546-647):*SidM*(546-647) | (*Hammond et al., 2014*) |
| Lyn$_{11}$-FRB-iRFP | piRFP-N1 | *LYN*(1-11):*MTOR*(2021–2113):iRFP | (*Hammond et al., 2014*) |
| mCherry-FKBP | pmCherry-C1 | mCherry:*FKBP1A*(3-108):[GGSA]$_4$GG | (*Hammond et al., 2014*) |
| mCherry-FKBP-SAC1 | pmCherry-C1 | mCherry:*FKBP1A*(3-108):[GGSA]$_4$GG:*SACM1L* | This study |
| mCherry-FKBP-SAC1$^{C389S}$ | pmCherry-C1 | mCherry:*FKBP1A*(3-108):[GGSA]$_4$GG:*SACM1L*$^{C389S}$ | This study |
| mCherry-SAC1-FKBP | pmCherry-C1 | mCherry:*SACM1L*:*FKBP1A*(3-108) | This study |
| mCherry-SAC1$^{C389S}$-FKBP | pmCherry-C1 | mCherry:*SACM1L*$^{C389S}$:*FKBP*(3-108) | This study |
| SAC1$^{\Delta452-587}$-FKBP-mCherry | pmCherry-N1 | *SACM1L*(1-451):*FKBP1A*(3-108):mCherry | This study |
| mCherry-FKBP-SAC1$^{\Delta TMD}$ | pmCherry-C1 | mCherry:*FKBP1A*(3-108):[GGSA]$_4$GG:*SACM1L*(1-521) | This study |
| mCherry-FKBP-SAC1-HLx2 | pmCherry-C1 | mCherry:*FKBP1A*(3-108):[GGSA]$_4$GG::*SACM1L*(1-520):[EAAAR]$_2$:*SACM1L*(521-587) | This study |
| mCherry-FKBP-SAC1-HLx4 | pmCherry-C1 | mCherry:*FKBP1A*(3-108):[GGSA]$_4$GG::*SACM1L*(1-520):[EAAAR]$_4$:*SACM1L*(521-587) | This study |
| mCherry-FKBP-SAC1-HLx6 | pmCherry-C1 | mCherry:*FKBP1A*(3-108):[GGSA]$_4$GG:SACM*1L*(1-520):[EAAAR]$_6$:*SACM1L*(521-587) | This study |
| mCherry-FKBP-SAC1-HLx8 | pmCherry-C1 | mCherry:*FKBP1A*(3-108):[GGSA]$_4$GG:*SACM1L*(1-520):[EAAAR]$_8$:*SACM1L*(521-587) | This study |
| mCherry-FKBP-SAC1-HLx10 | pmCherry-C1 | mCherry:*FKBP1A*(3-108):[GGSA]$_4$GG:*SACM1L*(1-520):[EAAAR]$_{10}$:*SACM1L*(521-587) | This study |

*Table 1 continued on next page*

*Table 1 continued*

| Plasmid | Backbone | Insert | Ref |
|---|---|---|---|
| mCherry-FKBP-SAC1$^{C389S}$-HLx8 | pmCherry-C1 | mCherry:*FKBP1A*(3-108):[GGSA]$_4$GG:*SACM1L*$^{C389S}$(1-520):[EAAAR]$_8$:*SACM1L*(521-587) | This study |
| mCherry | pmCherry-C1 | mCherry | (*Hammond et al., 2014*) |
| mCherry-SAC1 | pmCherry-C1 | mCherry:*SAC1ML* | (*Sohn et al., 2016*) |
| mCherry-SAC1-HLx8 | pmCherry-C1 | mCherryL*SACM1L*(1-520):[EAAAR]$_8$:*SACM1L*(521-587) | This study |
| mCherry-SAC1$^{C389S}$-HLx8 | pmCherry-C1 | mCherry:*SACM1L*$^{C389S}$(1-520):[EAAAR]$_8$:*SACM1L*(521-587) | This study |
| mTagBFP2-FKBP-CYB5A$^{tail}$ | pmTagBFP2-C1 | mTagBFP2:*FKBP1A*(3-108):[GGSA]$_4$GG:*CYB5A*(100-134) | This study |

DOI: https://doi.org/10.7554/eLife.35588.026

produced and sequenced with the GFP-11 forward primer to ensure correct in-frame addition of the cassettes.

## Fluorescence microscopy

For imaging, cells were placed in complete imaging medium consisting of Fluorobrite DMEM (ThermoFisher, A1896702), 10% heat-inactivated fetal bovine serum, 0.1% chemically-defined lipid supplement, 2 mM Glutamax (ThermoFisher 35050061) and 25 mM NaHEPES, pH 7.4. Where indicated, cells were stained with 1 µg/ml CellMask deep red in this medium for 3 min before exchanging for fresh media. For experiments in *Figure 3*, serum was omitted from the media. Cells were imaged on a Nikon Eclipse TiE inverted microscope using a 100x, plan apochromatic, 1.45 NA oil-immersion objective lens (Nikon). Excitation in either imaging mode was achieved using a dual fiber-coupled LUN-V 4-line laser launch with 405 nm (for TagBFP2), 488 nm (GFP), 561 nm (mCherry) or 640 nm (iRFP, CellMask Deep Red) laser lines. Cells were imaged in 2 ml imaging medium, or else 1.6 ml with the addition of 0.4 ml of imaging medium containing five-fold the final concentration of agonist or inhibitor during time-lapse acquisition, as indicated. A motorized stage (Nikon) was used to move between up to 16 consecutive positions in the dish for each time point with high precision. Acquisition was controlled using 'Elements' software (Nikon) and raw data including metadata were saved in 'nd2' format.

For confocal imaging, a Nikon A1R confocal scan head was used operating in resonant mode. 8 or 16 scans were integrated to improve signal to noise. To prevent cross-talk between channels, blue (425–475 nm) and yellow/orange (570–620 nm) fluorescence was acquired on a separate excitation scan to the green (500–550 nm) and far red (663–737 nm) channels. The confocal pinhole was set to 1.2x the size of the Airy disc of far-red fluors.

For TIRF imaging, a TIRF illuminator arm (Nikon) was used to deliver wide-field illumination at an acute angle. Emission for blue and yellow/orange (420–480 nm and 570–620 nm) along with green and far red/infrared (505–550 nm and 650–850 nm) was acquired using dual-pass filters (Chroma), mounted in adjacent positions in a Lamda 10–2 filter wheel (Sutter). Images were collected with a Zyla 5.5 sCMOS camera (Andor).

## Data analysis

Images were imported into the open-access image analysis platform Fiji (Schindelin et al. 2012) for analysis. Quantitative image analysis of multiple positions on the dish was performed in parallel by first assembling the images from each position into a single montage, and then generating regions of interest (ROI) of each cell to be analyzed.

For analysis of fluorescence intensity changes at the ER or PM from confocal data, images of the ER (using expressed iRFP-Sec61β) or the PM (using CellMask deep red) were used to generate a binary mask through à trous wavelet decomposition (Olivo-Marin 2002) as described previously (*Hammond et al., 2014*), using an automated custom-written ImageJ macro. Firstly, each image in the montage is normalized to the mean pixel intensity of the cell ROIs, to adjust for differences in expression level. A smoothing filter with a Gaussian approximation [1/16, 1/4, 3/8, 1/4, 1/16] is applied to the images to be used as the mask over three progressively larger length scales (s)

generating four images: $I_0$ - $I_3$ (the original plus the three smoothed). These images $I_s$ are used to compute the wavelets of these images $W_s = I_s - I_{s-1}$. After eliminating negative pixel values, these wavelet images $W_s$ are then multiplied together, resulting in a filtered image. A binary mask is then computed by thresholding at 3-fold the standard deviation of the filtered image. This mask is used to measure the fluorescence intensity in the ER or PM for the P4M-labelled images, after normalizing these to the mean pixel intensity of the cell ROIs (which again adjusts for differing expression levels between cells).

For analysis of the change in fluorescence intensity from TIRF data, images were background subtracted and then the mean fluorescence intensity was measured for total cell ROI throughout the time lapse. Images at time t were normalized to the mean ROI intensity pre-stimulation, that is $F_t/F_{pre}$.

To calculate the 'MCS index', images were recorded of each cell corresponding to expression of a GFP test protein, mCherry-MAPPER (as an MCS marker) and iRFP-Sec61β (as an ER marker). For each of these channels, pixel intensity was normalized to the maximum pixel intensity in a given cell ROI, giving a range from 0 to 1. This results in normalized images $I_{MCS}$, $I_{ER}$ and $I_{test}$. The differences between test protein and the markers is then computed, that is $dev_{MCS} = |I_{MCS} - I_{test}|$ and $dev_{ER} = |I_{ER} - I_{test}|$. Finally, MCS index was computed as the difference between ER and MCS deviations, MCS index = $dev_{ER} - dev_{MCS}$. Therefore, for MCS-localized test proteins, the deviation from ER was large and the deviation from MCS small, so MCS index is larger and positive. For ER-localized test proteins, deviation from MCS is large and from ER is small, giving smaller and more negative values.

Data were exported into Prism 7 (Graphpad) for graphing and statistical analysis. Data were subject to D'Agostino and Pearson normality tests which showed significant deviation of the data from Guassian distributions for every data set. Therefore, we used the non-parametric Kruskall-Wallis test with appropriate post-hoc tests for multiple comparisons; for two-way ANOVA, data were transformed with the natural logarithm to approximate a normal distribution.

Representative images were selected on the basis of the cells having good signal to noise and a metric (such as MCS index or change in $F_t/F_{pre}$) close to the median. Linear adjustments to the displayed dynamic range were performed for clarity. For images showing $F_t/F_{pre}$, an average intensity projection of the pre-bleach frames was used to normalize the entire time-lapse; pixels outside the ROI were set to zero (since background was divided by background and has a signal ~1). All images are thus shown at the same intensity scale of 0.3–1.1 and are directly comparable.

## Acknowledgements

We thank Jen Liou (UTSW), Pietro DeCamilli (Yale School of Medicine), Tom Rapoport (Harvard Medical School), Michael Davidson (Florida State University), Timothy Sanders (University of Pittsburgh) and Tamas Balla (NIH) for generously sharing plasmids. We are indebted to Aarika Yates and the Unified Flow Core (Department of Immunology, University of Pittsburgh School of Medicine) for assistance with fluorescence activated cell sorting. This work was supported by National Institutes of Health grant 1R35GM119412-01 (to GRVH).

## Additional information

### Funding

| Funder | Grant reference number | Author |
| --- | --- | --- |
| National Institutes of Health | 1R35GM119412-01 | James P Zewe<br>Rachel C Wills<br>Sahana Sangappa<br>Brady D Goulden<br>Gerald RV Hammond |

The funders had no role in study design, data collection and interpretation, or the decision to submit the work for publication.

## Author contributions

James P Zewe, Conceptualization, Resources, Data curation, Formal analysis, Supervision, Investigation, Methodology, Writing—review and editing; Rachel C Wills, Resources, Formal analysis, Investigation, Writing—review and editing; Sahana Sangappa, Formal analysis, Investigation; Brady D Goulden, Resources, Formal analysis, Investigation; Gerald RV Hammond, Conceptualization, Resources, Data curation, Software, Formal analysis, Supervision, Funding acquisition, Investigation, Methodology, Writing—original draft, Project administration, Writing—review and editing

## Author ORCIDs

James P Zewe (iD) http://orcid.org/0000-0001-5086-9052
Rachel C Wills (iD) http://orcid.org/0000-0003-2161-9235
Brady D Goulden (iD) http://orcid.org/0000-0002-1954-2091
Gerald RV Hammond (iD) http://orcid.org/0000-0002-6660-3272

## Decision letter and Author response

Decision letter https://doi.org/10.7554/eLife.35588.032
Author response https://doi.org/10.7554/eLife.35588.033

## Additional files

### Supplementary files

• Transparent reporting form
DOI: https://doi.org/10.7554/eLife.35588.027

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
