## [Decision Letter]

Thank you for submitting your work entitled "SAC1 Degrades PtdIns4*P* in the Endoplasmic Reticulum to Maintain a Steep Chemical Gradient with Donor Membranes" for consideration by *eLife*. Your article has been reviewed by three peer reviewers, and the evaluation has been overseen by a Reviewing Editor and a Senior Editor. The reviewers have opted to remain anonymous.

The reviewers commend you for your rigorous study of Sac1 localization and activity in vivo. However, the discussion between the reviewers led to the conclusion that the study essentially affirms the prevailing view of the field that Sac1 acts on substrate that is presented on ER (in "cis"). In addition, the reviewers concluded that the experiments do not formally exclude a mechanism where Sac1 acts on substrate in trans, perhaps in conditions not examined in your study.

*Reviewer #1:*

This manuscript by Zewe et al. describes a study of the mode by which the Sac1 PIP phosphatase functions. In 2011 (Stefan et al.), the Emr group proposed that Sac1 dephosphorylates its substrate in 'trans' where the Sac1 catalytic domain is tethered to the ER membrane and dephosphorylates PI4P in the plasma membrane, a view also suggested by Dickson et al., 2016. Contrary to this model, Mesmin et al., (2013) and Cai et al., (2014) suggested that Sac1 dephosphorylates PI4P in the same membrane, that is, in cis. This is an important distinction for understanding the chemical forces that are harnessed for trafficking lipids at organelle membrane contact sites.

Technically, this is a superb study. The potential for it lies in the claim that it has definitively settled the cis versus trans functional mode question. I note that the Mesmin et al., (2013) and Cai et al., (2014) papers strongly implied a cis mode for Sac1 and this view, it is fair to say, is currently well accepted by most in the field. It is possible that Sac1 can function in either a cis or trans mode in vivo. Thus, the main question is, does this study definitely settle the question once and for all. On this, the study falls short.

1) The authors suggest that an enrichment of Sac1 at contact sites is requisite for trans activity. I don't follow the logic. The trans mode requires that Sac1 localize to a contact site but does not require that it be enriched at these structures. The cis model only demands that Sac1 localize to the ER, which has been firmly established previously. The analysis clearly shows that Sac1 is not enriched at contact sites. I agree with the authors (subsection “5.3. “cis” and “trans” activity of SAC1 in cells”) that this observation does not definitively rule out or support either functional mode.

2) Using the FRB-FKBP system to artificially tether various portions of Sac1 to cellular membranes via the N- or C-terminus, and to artificially tether ER and PM to each other, the authors report that trans activity is only observed when the Sac1 tether is longer than that of the native enzyme. The results are consistent with the cis interpretation, but the extent to which contact sites generated in this manner relate to the native situation in cells is not established. More importantly, the approach imposes an entirely new set of constraints on Sac1 and the degree to which the study is informative for understanding native Sac1 is not firmly established.

*Reviewer #2:*

This study provides a definitive mechanism on how PtdIns4P-phosphatase Sac1 works in a cellular context. This paper is clear and technically impressive. The evidence in the paper is based on pharmacological and genetic approaches that are convincing and often elegant. It demonstrates that Sac1 is not capable of acting in a "trans" configuration in living cells. There was still uncertainty about the protein activity mode, that is, in "cis" or "trans" configuration, as the best-established models for either mode of action were based primarily on in vitro observations. Nevertheless, a body of evidence suggested that Sac1 operates in "cis" because it is the only way to explain how a family of lipid transfer proteins (OSBPs) uses PtdIns4P by counter-exchange and hydrolysis in "cis" to actively transport other cargo lipids between membranes, against their concentration gradient. In other words, a "phosphoinositide-motive force" is at work because the membranes that synthesize PtdIns4P are different from those that degrade it. That is why an action of Sac1 in "trans" seemed unlikely. Now, this paper demonstrates this formally.

I have one suggestion that could reinforce the manuscript prior to its publication. Indeed, some papers still maintain that OSBPs' activity is independent of Sac1. Is it then possible to further verify/refute that? The authors could measure plasma membrane enrichment of phosphatidylserine, by using a specific probe, such as GFP-evt2-2XPH or LactC2 (as described in Chung et al., 2015) depending on the activity of Sac1. Does a Sac1 inhibitor, or Sac1-HLx8 overexpression, alter the PS flux into the plasma membrane? This control would be valuable given the controversies in the field.

*Reviewer #3:*

In this study, Zewe et al. showed that SAC1 is localized in the ER but is not enriched at ER-PM MCS in both resting and receptor-stimulated cells. In addition, they found that adding a long (> 6 nm) helical linker near the transmembrane region enables SAC1 to efficiently dephosphorylate PI4P at the PM when SAC1 is recruited to ER-PM MCS. Interestingly, they found that only SAC1 with a long linker, but not the wildtype SAC1, confers the trans activity at artificially induced ER-PM MCS. Overall, their results showed that a significant trans activity of SAC1 can only be observed under manipulated conditions (i.e. with an inserted long linker plus forced recruitment to ER-PM MCS or induction of artificial ER-PM MCS), supporting previous structural studies that SAC1 is unlikely to act in trans at MCS (Cai et al., 2014). This is a nice study providing evidence derived from sophisticated studies in living cells which strongly suggests SAC1 mainly acts in cis. Nevertheless, there are several weaknesses which should be addressed before the publication.

1) Throughout this study, the authors relied on a single imaging probe P4M and its derivative P4Mx2 for their relative measurements of PI4P. The authors should use other imaging probes for PI4P, such as N-PH-ORP5, N-PH-ORP8L, and P4C_sidC (Chung et al., 2015 and Dong et al., 2016) to validate their results.

2) Subsection “5.1. Evidence for a “cis” acting SAC1” and Figure 1: The authors stated that mammalian cells do not survive in the absence of SAC1. They neglected to cite a recent work of the De Camilli Group (Dong et al., 2016) which reported changes in PI4P in various subcellular compartments, monitored by 3 different imaging probes, in HeLa cells knockout of SAC1 by CRISPR/Cas9. Dong et al. observed a strong increase in PI4P at the PM and endosomes in SAC1 knockout cells, whereas the authors observed an increase in PI4P in the ER following SAC1 inhibition as shown in Figure 1. The authors should address these discrepancies and examine PI4P in the ER using SAC1 knockout cells.

3) Figure 1 and the first few lines of subsection “5.1. Evidence for a “cis” acting SAC1”: With the existence of lipid transfer proteins, acute inhibition of SAC1 should lead to increase in PI4P in both donor and acceptor membranes, but the kinetics of the increase may differ depending on whether SAC1 acts in cis or in trans. The authors should monitor the kinetics of PI4P increases at both the ER and the PM following acute inhibition of SAC1. Various SAC1 constructs and rapamycin-based acute recruitment approach used in Figure 4, Figure 5 and Figure 6 can be used as controls to distinguish cis-acting and trans-acting SAC1.

---

## [Author Response]

We are submitting the manuscript again, with extensive revisions and new experimental data that address the reviewers’ comments. Specifically, we believe our new data addresses the overarching criticism that our data do not rule out the potential for “trans” activity of SAC1. We believe the new data shows that this cannot be the case.

Reviewer #1:[…] Technically, this is a superb study. The potential for it lies in the claim that it has definitively settled the cis versus trans functional mode question. I note that the Mesmin et al., (2013) and Cai et al., (2014) papers strongly implied a cis mode for Sac1 and this view, it is fair to say, is currently well accepted by most in the field. It is possible that Sac1 can function in either a cis or trans mode in vivo. Thus, the main question is, does this study definitely settle the question once and for all. On this, the study falls short.1) The authors suggest that an enrichment of Sac1 at contact sites is requisite for trans activity. I don't follow the logic. The trans mode requires that Sac1 localize to a contact site but does not require that it be enriched at these structures. The cis model only demands that Sac1 localize to the ER, which has been firmly established previously. The analysis clearly shows that Sac1 is not enriched at contact sites. I agree with the authors (subsection “5.3. “cis” and “trans” activity of SAC1 in cells”) that this observation does not definitively rule out or support either functional mode.

We are in complete agreement with the reviewer here. The reason we believed it important to detect the enrichment (or lack thereof) of SAC1 at MCS was because Dickson et al., (2016) proposed that modulating SAC1 localization at contact sites is an important mechanism of modulating “trans” activity.

Our data demonstrate that this is in fact not the case; “the limited and unchanging localization of SAC1 at MCS therefore appears co-incidental with its well-known distribution throughout the ER” (added to the end of this Results section).

“We also made the need to contrast with Dickson et al., 2016 more apparent in a new, more succinct opening paragraph of this section: “Dynamic recruitment of SAC1 to ER-PM MCS was recently proposed as a mechanism to modulate “trans” activity of the enzyme (Dickson et al., 2016). Although this is not inconsistent with the firmly established localization of SAC1 throughout the ER (Rohde et al., 2003; Nemoto et al., 2000), most proteins known to function at MCS are enriched at them too (Gatta and Levine, 2017). Therefore, a clue as to SAC1’s preferred mode of activity may be gleaned from a careful analysis of its enrichment (or not) at MCS.” (Subsection “SAC1 does not enrich at ER-PM MCS”.)

2) Using the FRB-FKBP system to artificially tether various portions of Sac1 to cellular membranes via the N- or C-terminus, and to artificially tether ER and PM to each other, the authors report that trans activity is only observed when the Sac1 tether is longer than that of the native enzyme. The results are consistent with the cis interpretation, but the extent to which contact sites generated in this manner relate to the native situation in cells is not established.

This is indeed a crucial point, and we have added new experimental data (Figure 6) to address this issue. Here we show that our PM and ER-targeted proteins (Figure 6) induce contact sites that have a narrower gap than those induced by native proteins, because these native proteins are squeezed out to the periphery of the induced MCS. It follows that since we report in Figure 6 that an extended linker is still required to confer “trans” activity at these narrow MCS, SAC1 could not possibly reach across the gap at native MCS.

We describe these new results as follows: “For a more acute interrogation of “trans” activity, we next induced dimerization between PM targeted FRB and an ER-localized FKBP (fused to the membrane anchor of cytochrome B5A; Komatsu et al., 2010) to generate new ER-PM MCS (Figure 6). These induced contact sites formed in under a minute at existing ER-PM MCS marked by GFP-E-Syt2 or GFPORP5 (Figure 6). In fact, as soon as they formed, the induced contact sites forced E-Syt2 and ORP5 to the periphery, indicating that the newly formed contacts were too narrow to accommodate native contact site proteins (Figure 6). This “squeezing out” of the contact site markers was dependent on ER-PM bridging, since GFP-Sec61β, which is not constrained to the PM-facing side of the ER, was not pushed to the periphery (Figure 6).” (Subsection ““Cis” and “trans” activity of SAC1 in cells”.)

More importantly, the approach imposes an entirely new set of constraints on Sac1 and the degree to which the study is informative for understanding native Sac1 is not firmly established.

The data in the original Figure 6 was included to address this crucial point. In this figure, SAC1 is overexpressed, but not directly conjugated to the dimerization system; MCS are instead induced by separate ER and PM-targeted halves of the FRB/FKBP system (see Figure 6). We have now strengthened this conclusion with the new data presented in Figure 6, where we instead induced expanded MCS by over-expressing the native MCS protein E-Syt2:

“To test for the propensity of SAC1 to act in “trans” at ER-PM MCS under unconstrained conditions, we capitalized on the observation that extended synaptotagmins expand MCS when over-expressed (Giordano et al.; Fernández-Busnadiego et al., 2015; Figure 6). These expanded contact sites would be expected to present more endogenous “trans” acting SAC1 to the plasma membrane, thus depleting PM PtdIns4P (Dickson et al., 2016). However, we found that expanded ER-PM MCS induced by E-Syt2 overexpression had no effect on PtdIns4P biosensor localization at the PM (Figure 6). In contrast, extensive depletion was observed after over-expression of ORP5, which is known to deplete PtdIns4P by transferring it back to the ER (Chung et al., 2015; Sohn et al., 2016).” (Subsection ““Cis” and “trans” activity of SAC1 in cells”.)

Reviewer #2:

[…] I have one suggestion that could reinforce the manuscript prior to its publication. Indeed, some papers still maintain that OSBPs' activity is independent of Sac1. Is it then possible to further verify/refute that? The authors could measure plasma membrane enrichment of phosphatidylserine, by using a specific probe, such as GFP-evt2-2XPH or LactC2 (as described in Chung et al., 2015) depending on the activity of Sac1. Does a Sac1 inhibitor, or Sac1-HLx8 overexpression, alter the PS flux into the plasma membrane? This control would be valuable given the controversies in the field.

This is an excellent suggestion, especially given the Balla lab’s recent demonstration that inhibited PtdIns4P synthesis in the PM leads to decreased PtdSer levels (Sohn et al., 2016). We attempted two experiments as suggested by the reviewer. Firstly, we followed the PM enrichment of the Lact-C2 PtdSer probe during peroxide treatment. This indeed led to a small but significant decrease in PM signal (Author response image 1). We also attempted to observe increases in ER-localized Lact-C2 as evidence of “back-traffic” of PtdSer due to the now collapsed PtdIns4P gradient. This led to a significant accumulation of ER-signal, but only if we over-expressed ORP5 to accelerate PtdSer traffic (Author response image 1). In contrast, accumulation of P4M in these cells was unaffected by ORP5 over-expression (Author response image 1). Collectively, the impact on the PtdSer biosensor was less dramatic than with the PtdIns4P biosensors. We believe the explanation likely stems from the fact that multiple OSBP-related proteins participate in the ER accumulation of PtdIns4P, but only ORP5 and ORP8 will contribute to PtdSer traffic from the PM. Hence the dramatic effects on PM PtdSer reported by Sohn et al. likely take much longer to develop.

**Author response image 1. respfig1:** Effects of SAC1 inhibition (with 500 µM peroxide) on PtdSer probe Lact-C2. (**A**) COS-7 transfected with mCherry-Lact-C2 were treated with vehicle or 500 µM DMSO during imaging by TIRF microscopy. Data are means ± s.e. of 32 cells from two independent experiments. P value is from a two-way repeated measures ANOVA. (**B**) mCherry-Lact-C2 and (**C**) TagBFP2-P4M were co-transfected into the same cells with iRFP-Sec61β (as an ER marker) and either GFP-ORP5 or GFP-calreticulin as control. Cells were imaged by confocal microscopy and assimilation of the probes fluorescence in the ER was measured. Data are means ± s.e. of 32-35 cells from three independent experiments. P values are derived from Sidak’s multiple comparison test after two-way repeated measures ANOVA (P < 10^–4^ for both).

The reviewers may feel differently, but we felt that these experiments were tangential to the main thrust of our story, so for the time being we have elected not to include these data in the manuscript.

Reviewer #3:

In this study, Zewe et al. showed that SAC1 is localized in the ER but is not enriched at ER-PM MCS in both resting and receptor-stimulated cells. In addition, they found that adding a long (> 6 nm) helical linker near the transmembrane region enables SAC1 to efficiently dephosphorylate PI4P at the PM when SAC1 is recruited to ER-PM MCS. Interestingly, they found that only SAC1 with a long linker, but not the wildtype SAC1, confers the trans activity at artificially induced ER-PM MCS. Overall, their results showed that a significant trans activity of SAC1 can only be observed under manipulated conditions (i.e. with an inserted long linker plus forced recruitment to ER-PM MCS or induction of artificial ER-PM MCS), supporting previous structural studies that SAC1 is unlikely to act in trans at MCS (Cai et al., 2014). This is a nice study providing evidence derived from sophisticated studies in living cells which strongly suggests SAC1 mainly acts in cis. Nevertheless, there are several weaknesses which should be addressed before the publication.1) Throughout this study, the authors relied on a single imaging probe P4M and its derivative P4Mx2 for their relative measurements of PI4P. The authors should use other imaging probes for PI4P, such as N-PH-ORP5, N-PH-ORP8L, and P4C_sidC (Chung et al., 2015 and Dong et al., 2016) to validate their results.

We used GFP-P4M as we believe (and have published, Hammond at al., 2014 and Hammond and Balla, 2015 as cited in the manuscript) that this is the only fully-characterized, un-biased probe for PtdIns4P in mammalian cells. We did however include new data showing that the ectopic pool of PtdIns4P formed in the ER is indeed detected with a second high affinity binding domain, P4C (new data shown in Figure 1). On the other hand, we did not believe it was worth repeating the experiments elsewhere in the paper where PM PtdIns4P was considered, since specificity of P4M has been so thoroughly characterized at this compartment previously.

**Author response image 2. respfig2:** Dissociation of ePH-ORP5 and -ORP8 by PtdIns4*P* or PtdIns(4,5)*P*_2_ depletion. COS-7 cells were transfected with GFP-tagged ePH probes, Lyn_11_-FRB-iRFP recruiter and mCherry-FKBP-SAC1^∆TMD^ or -INPP5E as indicated, and imaged by TIRF microscopy. Recruitment of enzymes was induced by rapamycin addition at time 0. Data are means ± s.e. of 6-10 cells from a single experiment.

To the reviewer’s suggestion of using Nterminal domains of ORP8 and ORP5 as PtdIns4P biosensors, recent data has proven these to be flawed probes. They bind – concurrently via two binding sites – both PtdIns4P and PtdIns(4,5)P_2_. This has been published by Ghai et al., 2017 and a paper is in press currently from the Balla lab extending these findings (that we have been collaborators on). To make sure the reviewers do not have to take our word for it, we also include an experiment in Author response image 2 showing dissociation of “extended PH” domains (ePH) from both ORP5 and ORP8 after depletion of either PtdIns4P or PtdIns(4,5)P_2_ with recruitable SAC1^∆TMD^ or INPP5E.

2) Subsection “5.1. Evidence for a “cis” acting SAC1” and Figure 1: the authors stated that mammalian cells do not survive in the absence of SAC1. They neglected to cite a recent work of the De Camilli Group (Dong et al., 2016) which reported changes in PI4P in various subcellular compartments, monitored by 3 different imaging probes, in HeLa cells knockout of SAC1 by CRISPR/Cas9. Dong et al. observed a strong increase in PI4P at the PM and endosomes in SAC1 knockout cells, whereas the authors observed an increase in PI4P in the ER following SAC1 inhibition as shown in Figure 1. The authors should address these discrepancies and examine PI4P in the ER using SAC1 knockout cells.

We apologize for this oversight – indeed, this study from the DeCamilli group is centrally relevant to our paper. We now cite this study in the opening paragraph of the Introduction, along with what we believe to be caveats to the interpretation of these data:

“Loss of SAC1 “trans” activity would cause accumulation of its substrate, PtdIns4P in membranes like the PM and Golgi where the lipid is synthesized, whereas loss of “cis” activity predicts accumulation of PtdIns4P in the ER (Figure 1). *Saccharomyces cerevisiae* with deletions of their Sac1 gene show 6-10 fold increases in PtdIns4P mass (Rivas et al.; Hughes et al., 2000; Guo et al., 1999), with PtdIns4P reported at both the PM (Roy and Levine, 2004; Stefan et al., 2011) and the ER (Roy and Levine, 2004; Tahirovic et al., 2005; Cai et al., 2014), depending on the probe used. RNAi of SAC1 in mammalian causes 1-2 fold accumulation of PtdIns4P (Cheong et al., 2010; Dickson et al., 2016; Goto et al., 2016), with accumulation reported in the ER (Cheong et al., 2010; Blagoveshchenskaya et al., 2008). On the other hand, acute knock-out of SAC1 in HeLa cells with CRISPR/Cas9 was reported to induce PtdIns4P accumulation on the PM and endosomes (Dong et al., 2016). However, these experiments are hard to interpret, since the SAC1 gene is essential to the survival of single mammalian cells (Blomen et al., 2015; Wang et al., 2015; Liu et al., 2008). Phenotypes in RNAi and knock-out experiments are therefore observed during the rundown of SAC1 protein levels before the cells die. The phenotype observed may thus be exquisitely sensitive to the precise amount of SAC1 protein remaining in the cell at the time of the experiment.” (Subsection “Evidence for a “cis” acting SAC1”).

It is important to note here that the Dong et al. study did not generate a true knock-out cell line; rather, the study relied on a short (72-hour) expression of Cas9 with an exon 5-targeted guide RNA under a puromycin selection cassette, which achieved a marked knock-down (but not knock-out) of SAC1 expression. We repeated this experiment at the reviewer’s suggestion, following the published protocol (Dong et al., 2016). Western blot of lysates from control or CRISPR/Cas9-treated cells using the same antiserum used by Dong et al. (a kind gift from Pietro De Camilli) showed many bands, owing to the fact that this is not an affinity-purified serum (Author response image 3). Nonetheless, we could see depletion of a band running slightly faster than the 70 kDa mark, consistent with 67 kDa SAC1 (Author response image 3), similar to what was previously published (Author response image 3; compare with Dong et al., Figure 2).

**Author response image 3. respfig3:** CRISPR/Cas9mediated acute kick-out of SAC1. (**A**) Western Blot of duplicate lysates from CRISPR/Cas9 plasmid transfected cells (selected with puromycin) after 72 h of knock down were probes with both SAC1 antisera and monoclonal anti-tubulin antibody DM1A. One sample was sonicated after lysis. The boxed region in blots is expanded in (**B**). Images of GFP-P4C (**C**) or GFPePH-ORP8 (**D**) from the 1st (1-25th percentile) or 4th (75-100th percentile) cohort of control or CRISPR/Cas9 plasmid transfected cells. The Box plots show mean and interquartile range, and the whiskers the list and 4th quartiles of 29 (**C**) or 25-32 (**D**) cells from a single experiment. Intensity at the PM was measured using a CellMask-derived mask as described in the manuscript.

In parallel, we wanted to see if we could reproduce the reported change in localization of GFP-P4C and ePH-ORP8. Whereas we did see a decreased PM accumulation of GFP-P4C that could be consistent with accumulated endosomal PtdIns4P (Author response image 3), we failed to observe the increased PM accumulation of GFP-ePH-ORP8, instead seeing a small decrease (Author response image 3). In addition, we noted that the localization of these probes in both control and “SAC1-KO” HeLa cells was quite variable. Using accumulation at the PM as a metric, and 1-25th percentile of P4C cells under either condition looked to have strong endosomal accumulation of probe (similar to KO in Dong et al), whereas the 75-100th percentile showed less endosomal accumulation and more PM signal (similar to control in Dong et al). Likewise, the lower quartile in ePH expressing cells showed very little PM localization (akin to control in Dong et al.), whereas the upper quartile showed strong PM localization (akin to SAC1 KO) under both conditions in our experiments.

In short, we were unable to reproduce a clear-cut difference between control and SAC1-KO cells. As we mention while discussing Dong et al. (see above), the discrepancy could be due to subtle differences in the timing of the experiments, leading to varying degrees of SAC1 protein rundown. It could also be due to differences in sampling between studies, in terms of which cells were imaged and measured. Either way, the amount of SAC1 protein present in a single cell used for these imaging studies cannot be known, even if knock down is effective across the population. Therefore, we conclude that these experiments are not informative as to SAC1’s preferred site of operation. Given this difficulty in interpretation, we prefer not to include these data in the manuscript.

3) Figure 1 and the first few lines of subsection “5.1. Evidence for a “cis” acting SAC1”: With the existence of lipid transfer proteins, acute inhibition of SAC1 should lead to increase in PI4P in both donor and acceptor membranes, but the kinetics of the increase may differ depending on whether SAC1 acts in cis or in trans. The authors should monitor the kinetics of PI4P increases at both the ER and the PM following acute inhibition of SAC1. Various SAC1 constructs and rapamycin-based acute recruitment approach used in Figure 4, Figure 5, and Figure 6 can be used as controls to distinguish cis-acting and trans-acting SAC1.

Monitoring PtdIns4*P* accumulation (or lack thereof) at the PM with our unbiased probes is not technically feasible, because the large accumulation at the ER depletes the pool of probe available for PM binding. Whilst the reviewer is quite correct, PtdIns4*P* accumulation at the PM is possible even after loss of “cis” activity, we argue that the appearance of PtdIns4*P* at the ER is cardinal for distinguishing “cis” vs “trans” activity. Changed levels at the PM in of itself is not a clear indicator of what mode the enzyme is acting in, without seeing ER accumulation (or lack thereof). Therefore, we feel that these experiments would be tangential to the study.